# Human hippocampal responses to network intracranial stimulation vary with theta phase

Sarah M Lurie[1]*, James E Kragel[2], Stephan U Schuele[3], Joel L Voss[2]

[1]Interdepartmental Neuroscience Program, Northwestern University, Chicago, United States; [2]Department of Neurology, University of Chicago, Chicago, United States; [3]Department of Neurology, Northwestern University, Chicago, United States

**Abstract** Hippocampal-dependent memory is thought to be supported by distinct connectivity states, with strong input to the hippocampus benefitting encoding and weak input benefitting retrieval. Previous research in rodents suggests that the hippocampal theta oscillation orchestrates the transition between these states, with opposite phase angles predicting minimal versus maximal input. We investigated whether this phase dependence exists in humans using network-targeted intracranial stimulation. Intracranial local field potentials were recorded from individuals with epilepsy undergoing medically necessary stereotactic electroencephalographic recording. In each subject, biphasic bipolar direct electrical stimulation was delivered to lateral temporal sites with demonstrated connectivity to hippocampus. Lateral temporal stimulation evoked ipsilateral hippocampal potentials with distinct early and late components. Using evoked component amplitude to measure functional connectivity, we assessed whether the phase of hippocampal theta predicted relatively high versus low connectivity. We observed an increase in the continuous phase–amplitude relationship selective to the early and late components of the response evoked by lateral temporal stimulation. The maximal difference in these evoked component amplitudes occurred across 180 degrees of separation in the hippocampal theta rhythm; that is, the greatest difference in component amplitude was observed when stimulation was delivered at theta peak versus trough. The pattern of theta-phase dependence observed for hippocampus was not identified for control locations. These findings demonstrate that hippocampal receptivity to input varies with theta phase, suggesting that theta phase reflects connectivity states of human hippocampal networks. These findings confirm a putative mechanism by which neural oscillations modulate human hippocampal function.

## Editor's evaluation

This paper provides important evidence in humans that the phase of neuronal oscillations is a key factor that varies functional processing in the brain and that phase information can be used to modulate activity in the memory network. The authors' findings of phase-sensitive stimulation effects are quite compelling because they use direct recordings from the human brain with advanced analytical tools, thus extending work in animals. This study will likely be of widespread interest, including for researchers interested in using brain stimulation to enhance human cognition as well as scientists who probe learning and memory in simpler animals.

*For correspondence: smr.lurie@gmail.com

Competing interest: The authors declare that no competing interests exist.

## Introduction

Episodic memory encoding and retrieval are thought to involve distinct hippocampal functional connectivity states (*Hasselmo and Stern, 2014*). During memory formation, fragments of episodic information are bound into coherent memory traces by the hippocampus. This process is thought to benefit from increased input via connectivity of entorhinal cortex to CA1 (*Hasselmo et al., 2002*; *Maass et al., 2014*; *Brankack et al., 1993*; *Kamondi et al., 1998*; *Fernández et al., 1999*) along with reduced recurrent hippocampal connectivity (*Hasselmo et al., 2002*). This connectivity pattern is thought to enhance the strength of incoming sensory signals while preventing interference from memory reactivation. Retrieval involves hippocampal-dependent reactivation (*Waldhauser et al., 2016*; *Tanaka et al., 2014*; *Gelbard-Sagiv et al., 2008*; *Tayler et al., 2013*; *Eichenbaum, 2000*), which is thought to benefit from the opposite functional connectivity pattern (*Hasselmo et al., 2002*; *Duncan et al., 2014*; *Montgomery and Buzsáki, 2007*).

In rodents, hippocampal theta oscillations orchestrate the transition between these distinct connectivity states. Rodent hippocampal theta synchrony supports memory formation and retrieval (*Markowska et al., 1995*; *Winson, 1978*), with different theta phases thought to support opposing functional connectivity patterns. For instance, tetanic electrical stimulation of hippocampal CA1 results in long-term potentiation when delivered at the stratum radiatum theta peak versus long-term depression when delivered at the trough (*Hölscher et al., 1997*; *Hyman et al., 2003*), supporting a theta-phase dependence of encoding readiness. In addition, disrupting the theta cycle with inhibitory stimulation locked to local peak versus trough has been shown to differentially impact encoding and retrieval in rodents (*Siegle and Wilson, 2014*).

It is currently unclear whether human hippocampus shows the phase dependence of receptivity to external input that has been identified in rodents, particularly given the numerous differences in human versus rodent hippocampal theta characteristics including the presence and functional relevance of high-power oscillatory bouts (*Goyal et al., 2020*; *Jacobs, 2014*; *Kahana et al., 1999*; *Raghavachari et al., 2001*). The goal of the present experiment was to test predictions of this theta-phase dependence model in humans. We recorded local field potentials (LFPs) from the hippocampus in patients undergoing intracranial electrophysiological recording via implanted depth electrodes as part of their clinical care. We applied direct electrical stimulation to hippocampal network sites in lateral temporal cortex with putative projections to entorhinal cortex and hippocampus (*Insausti et al., 1987*; *Catani and Thiebaut de Schotten, 2008*; *Zhong and Rockland, 2004*), and measured the hippocampal response to lateral temporal stimulation using well-characterized early and late evoked-potential components (*Matsumoto et al., 2004*; *Novitskaya et al., 2020*).

We hypothesized that if hippocampal receptivity to network input varies with the theta oscillation, then the hippocampal response to stimulation would differ according to hippocampal theta phase at the time of stimulation delivery. In rodents, fissural theta trough versus peak are the phases commonly related to maximal versus minimal receptivity to external input (e.g., *Hasselmo, 2005*). However, previous studies in rodents have reported disparate phase angles relating to maximal entorhinal–hippocampal transmission, likely due to differences across studies in the targeted hippocampal layer. Because the theta oscillation arises from interlaminar dipoles (*Goutagny et al., 2009*; *Kamondi et al., 1998*), its observed phase varies according to electrode depth. Studies in rodents have therefore reported maximal entorhinal input variously at the recorded hippocampal trough (from fissural recording, as in *Brankack et al., 1993*), peak (from pyramidal layer recording, as in *Douchamps et al., 2013*), and falling phases (from recording in variable layers; see *Siegle and Wilson, 2014*). Thus, we anticipated 180 degree separation in phases associated with maximal versus minimal receptivity in humans, without strong hypotheses for which phase angles would be associated with these states given the localization uncertainty in electrodes placed for clinical purposes in human subjects. We therefore tested this hypothesis by first analyzing continuous variation in amplitude based on theta phase following stimulation. We used a novel method to account for the phase dependence of amplitude values that occurs irrespective of stimulation. To test selectivity, we analyzed theta-phase dependence for non-hippocampal control locations in the amygdala and orbitofrontal cortex.

**Table 1.** Participant characteristics.

| Sex | Age | Hemisphere of electrodes | # hippocampal recording electrodes analyzed | Pulses delivered | Stimulation protocol |
|-----|-----|--------------------------|---------------------------------------------|------------------|----------------------|
| F | 28 | Right | 3 | 1721 | 0.5 and 1 Hz |
| F | 29 | Left | 3 | 1576 | 0.5 and 1 Hz |
| F | 30 | Left | 3 | 1170 | 0.5 and 1 Hz |
| F | 44 | Left | 4 | 241 | 0.5 Hz |
| F | 55 | Left | 1 | 2566 | 0.5 and 1 Hz |
| F | 47 | Left | 3 | 1036 | 0.5 and 1 Hz |
| M | 31 | Left | 4 | 1766 | 0.5 and 1 Hz |
| M | 29 | Right | 2 | 982 | 0.5 and 1 Hz |

# Results

## Participants and stimulation protocol

Data were collected from eight individuals with refractory epilepsy (two males; mean age ± standard deviation [SD]: 37 ± 10 years; range 28–55 years; *Table 1*) undergoing invasive electrophysiological monitoring as part of their inpatient clinical care at the Northwestern Memorial Hospital Comprehensive Epilepsy Center. All participants had stereotactic EEG (sEEG) depth macroelectrodes (Ad-Tech, Oak Creek, WI) implanted in hippocampus, amygdala, lateral temporal cortex, and orbitofrontal cortex, in addition to other regions. All stimulating electrodes were localized to lateral temporal cortex and adjacent white matter (*Figure 1a, b*; see *Figure 1—animation 1*).

In seven participants, the experimental stimulation protocol consisted of trains of single pulses (see Materials and methods: sEEG recording and stimulation) delivered at either 0.5 Hz (~60 pulses per train) or approximately 1 Hz (with an interpulse interval range of 1–1.25 s, jittered pseudorandomly, ~1200 pulses per train; interpulse interval and range of jitter values were selected to enable data analysis for a secondary experiment). These train types were alternated with approximately 2 min of rest between trains. In one participant, stimulation was delivered at 0.5 Hz only. The number of stimulation pulses delivered ranged from 241 to 2566 (*Table 1*). Stimulation did not elicit seizure or clinically significant afterdischarges in any participant.

## Hippocampal recordings showed narrowband oscillations within the 3–8 Hz theta range

To ensure the presence of theta activity in hippocampal LFPs, we assessed the power spectral densities of recordings taken during a rest period before the stimulation experiment. We estimated oscillatory power during this pre-stimulation period rather than during the experimental session to avoid contaminating the estimate with the evoked response to stimulation. We located narrowband oscillations by fitting a 1/*f* background distribution to the power spectrum (*Figure 1c*), subtracting this background, and estimating local peaks in the resultant curve (see Materials and methods: Analysis of narrowband theta activity). All analyzed hippocampal recording electrodes (as well as all amygdala and OFC control region recording electrodes) showed at least one narrowband oscillation in the 3–8 Hz range. Across hippocampal recording electrodes, multiple local peaks in oscillatory power were detected within the 3–8 Hz theta band (*Figure 1c*). Peaks where power was significantly greater than the 1/*f* background spectrum across electrodes were detected within the 3–8 Hz theta range at 5.3 and 6.3 Hz (*t*-test of corrected power versus 0, 5.3 Hz: $t(22) = 3.4$, $p = 0.003$; 6.3 Hz: $t(22) = 3.3$, $p = 0.003$). An additional peak was detected at 3.9 Hz; however, power at this lower frequency was only marginally above the 1/*f* background spectrum (3.9 Hz: $t(22) = 1.9$, $p = 0.07$). Peaks with significant power above background were also detected at 2.4, 11.0, and 17.7 Hz (*Figure 1c*; all $p < 0.05$ on *t*-test of corrected power versus 0).

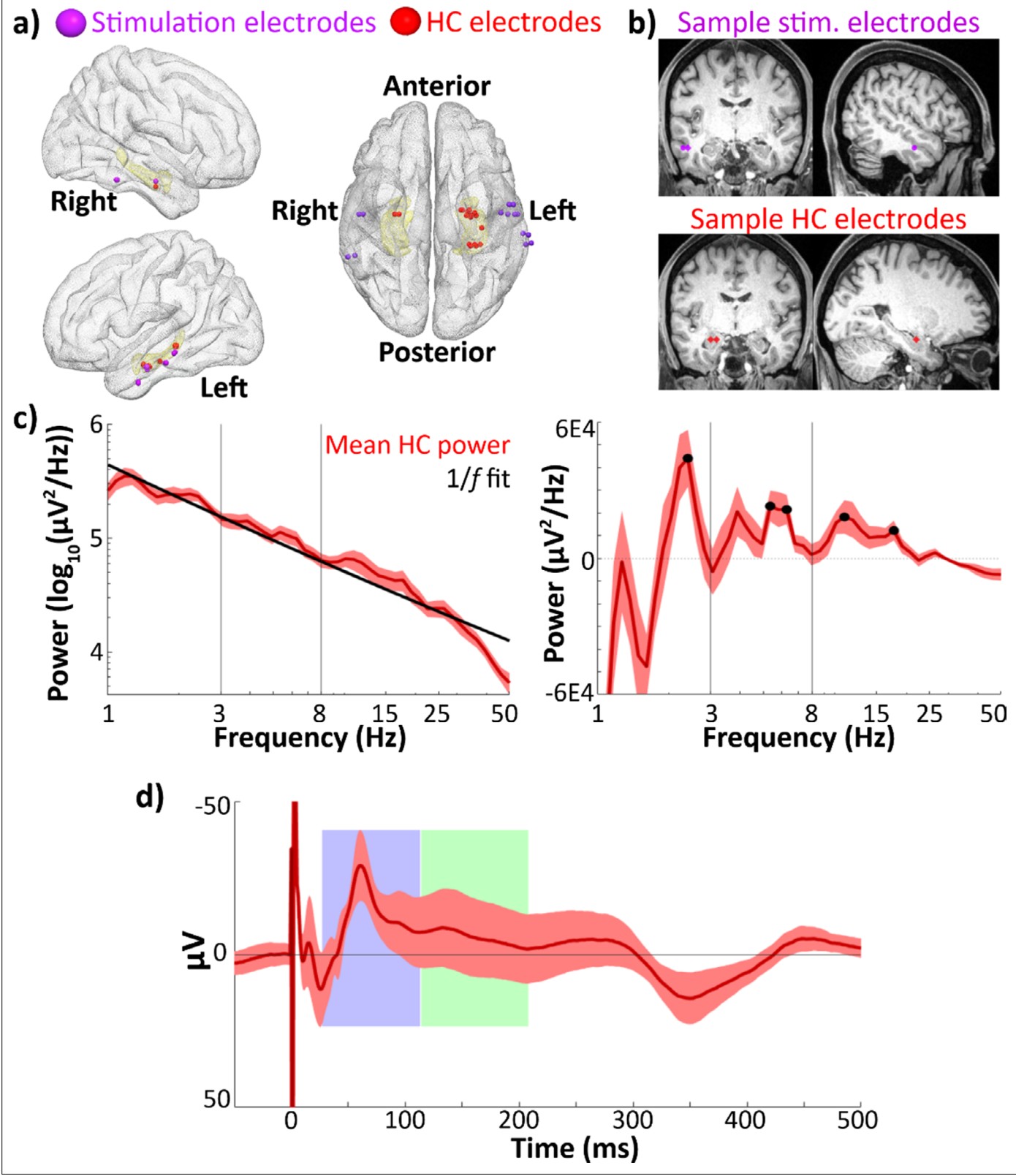

**Figure 1.** Hippocampal recordings and evoked response. (**a**) Group-level electrode localization for lateral temporal stimulating electrodes (*n* = 14) and hippocampal recording electrodes (*n* = 18) plotted on an MNI template (*Holmes et al., 1998*). Amygdala and hippocampus are highlighted in yellow. Electrodes are enlarged ~500% for visualization purposes. *Note*: Imaging was unavailable in one subject (see Materials and methods: Electrode localization). Hippocampal recording electrodes did not align to the template brain hippocampus in one subject and are not shown in this image. (**b**)

*Figure 1 continued on next page*

*Figure 1 continued*

Locations of lateral temporal stimulating electrodes (top) and recording electrodes in hippocampus (bottom) in one sample participant. (**c**) Mean power spectral densities across all hippocampal recording electrodes ($n = 23$). Power was assessed during a recorded pre-stimulation rest period. Left: Mean power spectral density (red) and estimated $1/f$ fit (black). Right: Mean $1/f$-corrected power spectral density across hippocampal recording electrodes. Significant peaks in oscillatory power marked in black. Error bars indicate ±1 standard error of the mean (SEM) across electrodes. (**d**) Phase-balanced grand average hippocampal evoked potential (EP) (i.e., 0°, 90°, 180°, and 270° phase bins contribute equally to the average) elicited by lateral temporal stimulation. Error bars indicate ±1 SEM across hippocampal recording electrodes. Identified early and late negative components are highlighted in blue and green. The *y*-axis shows negative values in the upwards direction to emphasize the negative-going EP components of interest.

The online version of this article includes the following video for figure 1:

**Figure 1—animation 1.** Rotating view of the group-level electrode localization shown in Figure 1a.

## Hippocampal evoked potentials showed characteristic early and late negative components

Hippocampal evoked-potential (EP) components were estimated to occur from 27 to 113 ms (early) and 114 to 208 ms (late) post-stimulation (*Figure 1d*; see Materials and methods: Quantification of hippocampal EPs). The observed component latencies were consistent with values reported by previous studies of human hippocampal response to direct electrical stimulation of polysynaptic afferents (e.g., *Kubota et al., 2013*; *Novitskaya et al., 2020*).

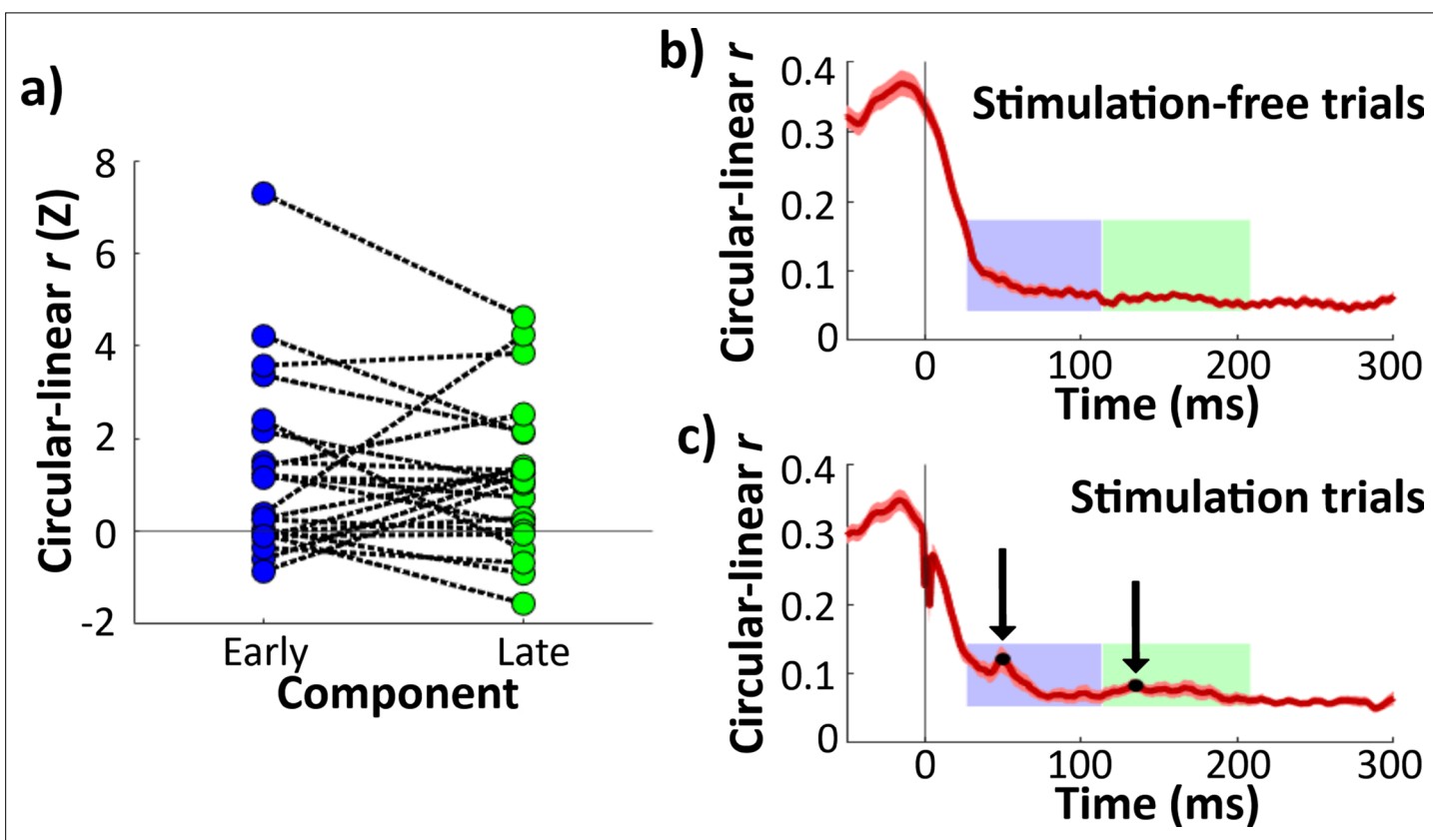

**Figure 2.** Continuous theta phase predicted response amplitude during early and late evoked-potential (EP) components. (**a**) Z-scored circular–linear correlation *r* for the relationship between hippocampal theta phase at stimulation onset and EP component amplitude. Each line represents one electrode. *Z*-scores are shown for early (left) and late (right) components. Horizontal line shows chance-level circularity. Both early and late components showed significant theta circularity (**p < 0.01). (**b**) Circular–linear *r* plotted for each timepoint for stimulation-free trials, Error bars indicate ±1 standard error of the mean (SEM) across hippocampal recording electrodes. Early and late component timecourses highlighted in blue and green for reference. (**c**) As (**b**), for stimulation trials. Local maxima denoted in black.

## Hippocampal EP amplitudes varied according to theta phase at stimulation onset

We performed circular–linear analyses to determine whether hippocampal receptivity to lateral temporal stimulation varied continuously (i.e., sinusoidally) with the phase of the theta oscillation. We first assessed whether amplitudes of the early and late EP components (*Figure 1d*) varied with theta phase at stimulation onset (see Materials and methods: Theta-phase estimation). For each electrode, we computed circular–linear correlation between hippocampal phase at stimulation onset and component amplitude across stimulation trials. Permutation testing was used to assess whether periodicity in evoked response amplitude was above chance. Both early and late component amplitudes were significantly predicted by theta phase at stimulation onset (*Figure 2a*. Mean $z$-score ± SD, early: $z = 1.3 ± 1.9$; late: $z = 1.2 ± 1.6$. $t$-test of $z$-scores versus 0, early: $t(22) = 3.3$, p = 0.003, Cohen's $d =$ 0.7; late: $t(22) = 3.7$, p = 0.001, Cohen's $d = 0.8$. Mean $r$ ± SD, early: $r = 0.080 ± 0.066$; late: $r = 0.076 ± 0.051$).

To assess whether the phase–amplitude relationship was appropriately captured by the early and late components, we performed an exploratory analysis of the phase–amplitude relationship across all timepoints in the peri-stimulation period. Additionally, to investigate whether the observed effect was caused by phase-dependent changes in the evoked response (as opposed to phase-dependent amplitude of the non-evoked, theta oscillatory component of the signal), we assessed the timecourse of the phase–amplitude relationship for the non-evoked response by performing the same analysis for phase-matched, stimulation-free trials (see Materials and methods: Comparison of stimulation trials to phase-matched stimulation-free trials).

In both stimulation-free and stimulation trials, the phase–amplitude relationship was strongest before and up to stimulation onset (*Figure 2b, c*). The asymmetric dropoff about $t = 0$ likely relates to the phase estimation method (i.e., amplitude before stimulation is more strongly predictive of phase because it contributes directly to the phase estimate; see *Appendix 1—figure 1b*). In stimulation-free trials, the phase–amplitude relationship timecourse following $t = 0$ was generally smooth and monotonic (*Figure 2b*). In contrast, stimulation trials exhibited local increases in circular–linear correlation (*Figure 2c*). Qualitatively, the local increases appeared to coincide with the early and late EP components (as plotted in *Figure 1d*). Using the same peak-finding method that identified peaks in the EP (see Materials and methods: Quantification of hippocampal EPs), we found peaks in the phase–amplitude relationship at +50 and +135 ms following stimulation onset, which aligns closely with the early and late EP components (+62 and +134 ms, respectively; *Figure 2c*). Stimulation trials also exhibited a sharp local decrease from approximately +0 to +5 ms following stimulation, likely due to phase-independent artifact during and immediately after the stimulation pulse (*Figure 2c*). These findings indicate that hippocampal theta phase at stimulation onset predicts its responsiveness to lateral temporal stimulation, and that these effects are well captured by analyses of characteristic early and late components of the hippocampal EP.

## Hippocampal EP amplitudes varied for theta peak versus trough

Given the previous rodent findings of maximal differences in receptivity to input at specific hippocampal theta-phase angles (e.g., trough versus peak; *Brankack et al., 1993*; *Hasselmo et al., 2002*), we tested whether the phase-dependent responsivity of the hippocampus to stimulation identified in the analyses above varied for specific theta-phase angles. We estimated theta phase at stimulation onset for each trial (see Materials and methods: Theta-phase estimation) and binned trials to 90° intervals, centered on peak, trough, rising, and falling phases. By taking the means within each bin, we obtained average peak, trough, falling, and rising angle stimulation trials for each electrode (*Figure 3a, c*).

As phase at a given timepoint predicts future amplitude by definition, when trials are sorted according to theta phase at stimulation, differences are expected in the post-stimulation signal simply owing to the ongoing theta oscillation. We therefore isolated the evoked response from this non-evoked oscillation in order to assess whether the evoked response itself varied according to stimulation phase. We estimated the non-evoked oscillation using phase-matched stimulation-free trials (see Materials and methods: Comparison of stimulation trials to phase-matched stimulation-free trials). We binned stimulation-free trials to 90° intervals using the same approach as for the stimulation trials. By taking the mean of stimulation-free trials within each bin, we estimated the non-evoked component of

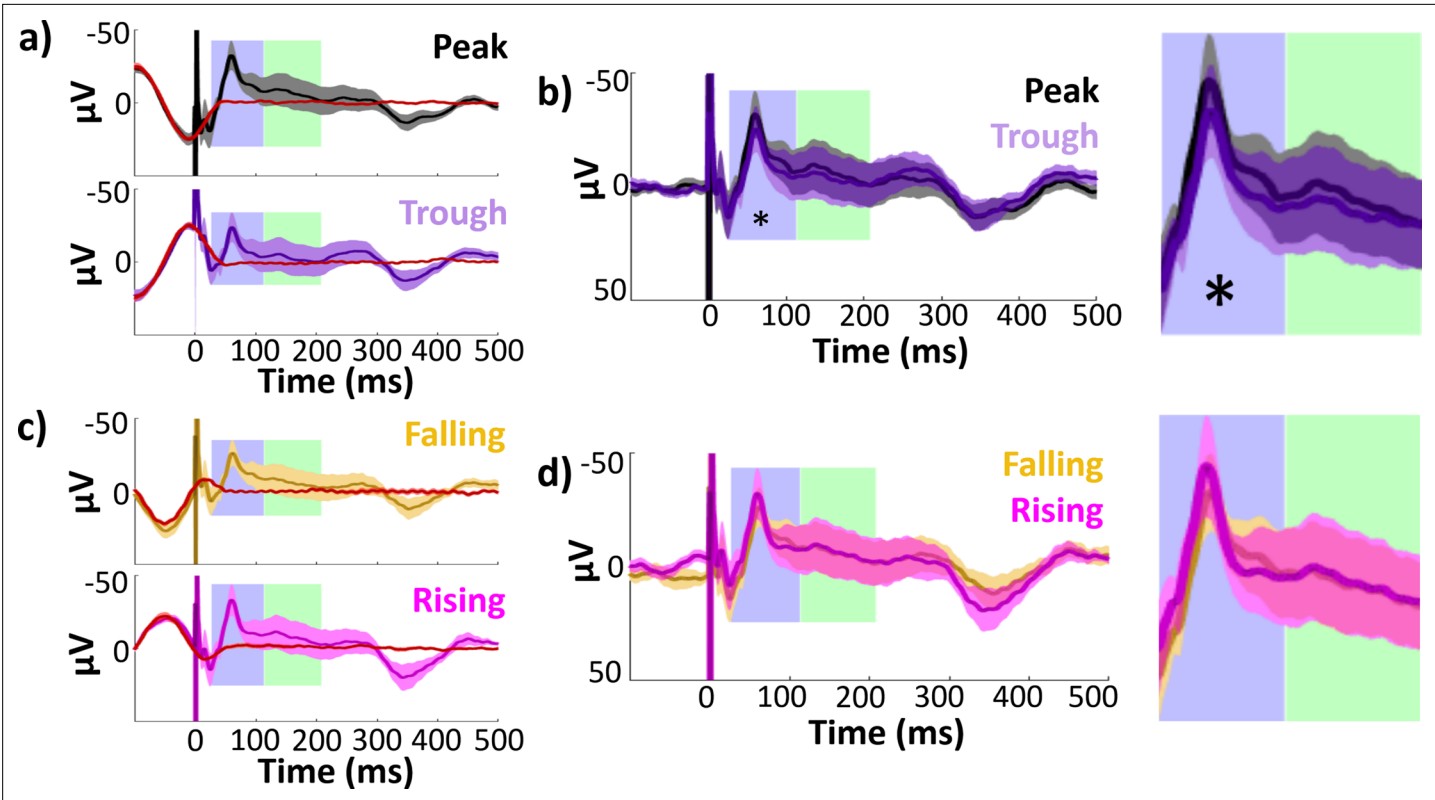

**Figure 3.** Hippocampal evoked potentials (EPs) and isolated responses binned according to theta phase at stimulation delivery. Early and late component timecourses highlighted in blue and green for reference. (**a**) Mean hippocampal EPs elicited by lateral temporal stimulation at theta peak (black) and trough (purple), alongside phase-matched stimulation-free trials (red; see Materials and methods: Comparison of stimulation trials to phase-matched stimulation-free trials). Error bars indicate ±1 standard error of the mean (SEM) across recording electrodes. Theta oscillation is visible around stimulation at $t = 0$. The isolated response was obtained by subtracting the mean phase-matched stimulation-free trial from the EP. (**b**) Isolated hippocampal evoked responses to peak and trough stimulation. The non-evoked oscillatory component is abolished in the isolated response. Asterisk indicates isolated response components with significant amplitude differences across phase bins at 180° intervals (i.e., peak versus trough or rising versus falling trials; *p ≤ 0.05). Left: Full timecourse. Right: Enlarged panels showing component timecourses. (**c, d**) As in (a) and (b), for stimulation at theta falling (yellow) versus rising (magenta) phase angles.

the peak and trough EPs for each electrode (red lines in *Figure 3a, c*). As expected, the non-evoked component was an oscillation with asymmetrical coherence dropoff around stimulation onset (see *Appendix 1—figure 1b*). Finally, to isolate the evoked response, we subtracted this non-evoked component from its corresponding EP. This procedure abolished pre-stimulus amplitude differences between peak versus trough and rising versus falling trials across hippocampal recording electrodes, indicating good removal of the ongoing oscillatory component (paired *t*-test on mean amplitudes −100 to 0 ms, peak versus trough: $t(22) = 0.47$, $p = 0.6$; rising versus falling: $t(22) = 1.4$, $p = 0.2$).

The early and late EP components showed significant amplitude differences following peak versus trough stimulation (*Figure 4a*. Paired *t*-test across hippocampal recording electrodes, early: $t(22) = −2.7$, $p = 0.01$, Cohen's $d = −0.2$; late: $t(22) = −2.6$, $p = 0.02$, Cohen's $d = −0.08$. Mean peak − trough amplitude difference ± SD, early: diff = −4.9 ± 8.5 µV, diff = −3.9 ± 7.2 µV). Isolation of the evoked response (*Figure 3b*) reduced the differences between peak and trough stimulation in the late component (*Figure 4b*. Paired *t*-test: $t(22) = −1.1$, $p = 0.3$, Cohen's $d = −0.04$). However, the peak versus trough effect in the early component persisted through the isolation procedure (*Figure 4b*. Paired *t*-test: $t(22) = −2.1$, $p = 0.05$, Cohen's $d = −0.2$; mean amplitude difference ± SD, early: diff = −4.8 ± 11 µV; late: diff = −2.2 ± 10 µV).

No difference in EP amplitude was observed for rising versus falling stimulation conditions (*Figure 3c*) for either early or late components (*Figure 4c*. Paired *t*-test across hippocampal recording electrodes, early: $t(22) = 0.45$, $p = 0.7$, Cohen's $d = 0.04$; late: $t(22) = 0.55$, $p = 0.6$, Cohen's $d = 0.03$; mean falling − rising amplitude difference ± SD: early: diff = 1.1 ± 11.3 µV; late: diff = 1.6 ± 14.0 µV).

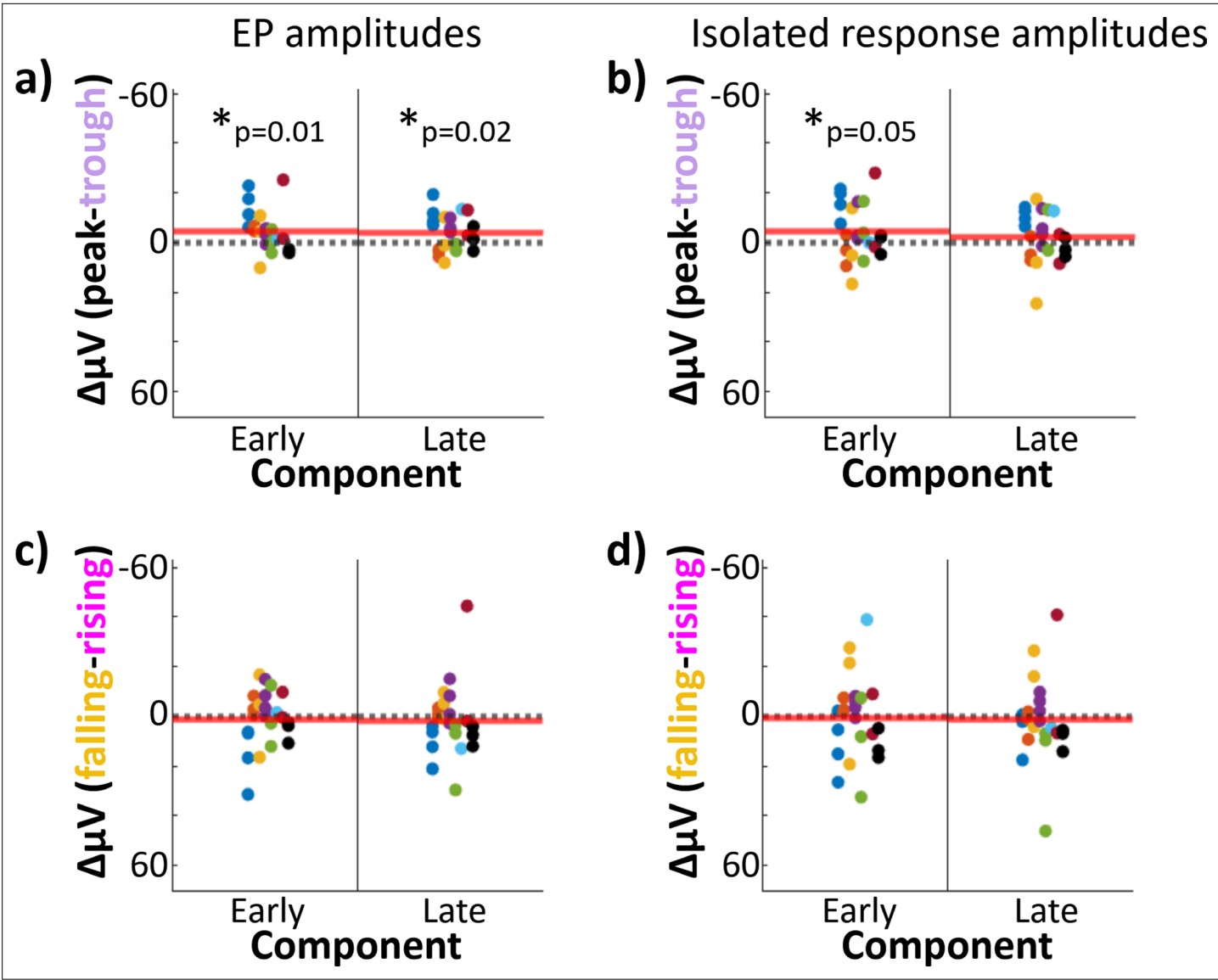

**Figure 4.** Stimulation at theta peak versus trough produced differences in early-component amplitude. Hippocampal component amplitudes plotted by theta phase of stimulation. Each dot represents one electrode; dot color indicates participant-of-origin. Red line indicates the mean difference across electrodes. Difference between hippocampal evoked-potential (EP) component amplitudes elicited by lateral temporal stimulation delivered at peak versus trough (**a**) and falling versus rising (**c**) phases. Asterisk indicates components with significant amplitude differences across phase bins at 180° intervals (i.e., peak versus trough or rising versus falling trials; *p ≤ 0.05). Left: Early component amplitude difference. Right: Late component amplitude difference. (**b, d**) As in (**a**) and (**c**), but for the isolated hippocampal response (i.e., EP minus phase-matched stimulation-free trials).

As was the case in the overall EP, no differences were observed in the isolated response for falling versus rising stimulation (*Figure 4d*. Paired *t*-test across hippocampal recording electrodes, early: $t(22) = 0.14$, $p = 0.9$, Cohen's $d = 0.02$; late: $t(22) = 0.41$, $p = 0.7$, Cohen's $d = 0.02$; mean falling − rising amplitude difference ± SD, early: diff = 0.52 ± 16.6 µV; late: diff = 1.4 ± 16.2 µV).

### Control regions did not show early-component-specific periodicity in the response to stimulation

To assess whether theta dependence of EP amplitude was specific to hippocampus, we performed the same analyses on data from recording electrodes in amygdala (n = 9) and orbitofrontal cortex (n = 22). Data from control regions were acquired concurrently with the hippocampal data described above.

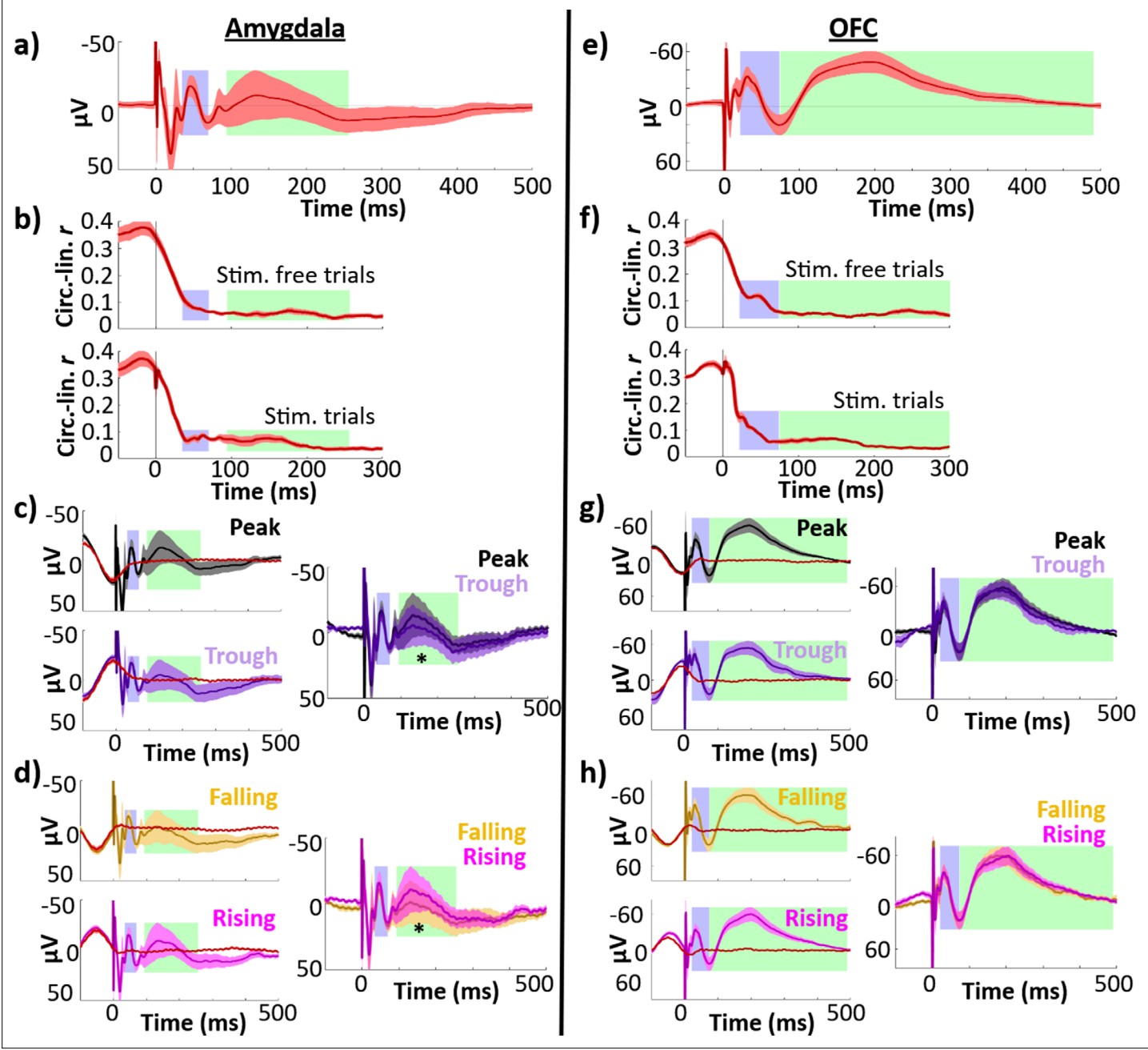

**Figure 5.** Control regions showed less temporal specificity in their phase–amplitude relationships and no difference in peak versus trough or falling versus rising early response amplitudes. Left column, amygdala (*n* = 9 electrodes); right column, OFC (*n* = 22 electrodes). Early (blue) and late (green) components are highlighted. Error bars indicate ±1 standard error of the mean (SEM) across electrodes. (**a**, **e**) Phase-balanced grand average evoked potential (EP). (**b**, **f**) Phase–amplitude circular–linear *r* plotted for each timepoint in the peri-stimulation period for stimulation trials (top) and stimulation-free trials (bottom). (**c**, **g**) Left: Mean EPs elicited by lateral temporal stimulation at local theta peak (black) and trough (purple), alongside phase-matched stimulation-free trials (red; see Materials and methods: Comparison of stimulation trials to phase-matched stimulation-free trials). The isolated response was obtained by subtracting the mean phase-matched stimulation-free trial from the EP. Right: Isolated evoked responses to peak and trough stimulation. Asterisk indicates isolated response components with significant amplitude differences across phase bins at 180° intervals (i.e., peak versus trough or rising versus falling trials; *$p \le 0.05$). (**d**, **h**) As in (**c**, **g**), for stimulation at local theta falling (yellow) versus rising (magenta) phase angles.

In both control regions, stimulation EPs had distinct early and late negative components (*Figure 5a, e*). In amygdala, components were estimated to span 35–70 ms (early) and 94–256 ms (late). In OFC, components spanned 22–74 ms (early) and 74–490 ms (late).

We performed circular–linear analyses to investigate whether control regions exhibited a continuous relationship between phase and EP amplitude. As we did in our analysis of the hippocampal phase–amplitude relationship (*Figure 2b, c*), we assessed the circular–linear correlation between phase and amplitude for each control region electrode at each timepoint of the EP. To control for the contribution of the non-evoked oscillatory signal to this relationship, we assessed the phase–amplitude relationship for both stimulation trials and stimulation-free trials. As was the case in the hippocampal EP, we observed the strongest phase–amplitude relationship before and up to stimulation onset (*Figure 5b, f*), with asymmetric dropoff about $t = 0$ likely due to the phase estimation method. In contrast to the hippocampal EP (see *Figure 2c*), both control regions showed more diffuse deviations in the phase–amplitude relationship. In particular, neither control region showed the clear, temporally specific increase in circular-linearity during the early component which was present in the hippocampal EP.

As in the analysis of the hippocampal response, we next binned EP trials according to phase at stimulation onset and compared component amplitudes across peak and trough trials. The amygdala EP showed significant amplitude differences across peak and trough trials in the late component (paired *t*-test: $t(8) = -4.1$, p = 0.003, Cohen's $d = -0.2$), with greater amplitude for peak trials (*Figure 5c*. Mean difference ± SD: $-5.5 \pm 4.0$ μV). There was no significant amplitude difference in the early component ($t(8) = 0.85$, p = 0.4, Cohen's $d = 0.3$). This effect persisted after we isolated the evoked response using phase-matched stimulation-free trials (late: $t(8) = -3.1$, p = 0.02, Cohen's $d = -0.2$; mean difference ± SD: $-5.3 \pm 5.2$ μV). Similarly, we observed amplitude differences for amygdala rising versus falling trials selective to the late component (*Figure 5d*. Early: $t(8) = -0.29$, p = 0.8, Cohen's $d = -0.06$; late: $t(8) = 2.7$, p = 0.03, Cohen's $d = 0.1$), which persisted through isolation of the evoked response (early: $t(8) = 1.2$, p = 0.3, Cohen's $d = 0.2$; late: $t(8) = 4.7$, p = 0.002, Cohen's $d = 0.2$).

We hypothesized that the phase dependence of the late component may have been driven by secondary transmission from the hippocampus. However, we did not find a consistent correlation between hippocampal EP and amygdala late component timing (see Appendix: Hippocampal EP timing did not predict amygdala late component latency).

In contrast, for orbitofrontal EPs, we found no differences in either peak versus trough or rising versus falling trial amplitude in either early or late components (*Figure 5g, h*. Peak versus trough, early: $t(21) = 0.51$, p = 0.6, Cohen's $d = 0.04$; late: $t(21) = -0.77$, p = 0.5, Cohen's $d = -0.1$. Rising versus falling, early: $t(21) = -0.59$, p = 0.6, Cohen's $d = -0.04$; late: $t(21) = -0.08$, p = 0.9, Cohen's $d = -0.007$). While our response-isolation procedure revealed a marginal effect of peak versus trough stimulation on the orbitofrontal early-component amplitude ($t(21) = 2.1$, p = 0.05, Cohen's $d = 0.07$), this effect may have been driven by poor performance of the sham-matching procedure for the orbitofrontal EP (see *Figure 5g, h*; we observe poor abolishment of the underlying theta oscillation for both trough and rising stimulation trials).

## Analysis of variability across subjects

As the previous analyses were performed across individual electrodes, we assessed whether the circular–linear and binned effects of theta phase on response component amplitude were present across subjects. For each subject, we obtained the mean z-scored circular–linear r across hippocampal electrodes for early and late components (as described for individual hippocampal electrodes in Materials and methods: Circular–linear analysis of theta phase and hippocampal response amplitude). Across subjects, we observed a positive phase–amplitude relationship in both early and late components (mean z-score ± SD, early: $z = 1.0 \pm 1.3$; late: $z = 1.1 \pm 1.1$). This effect was marginal across subjects in the early component and reached significance in the late component (t-test of z-scores versus 0, early: $t(7) = 2.3$, p = 0.06, Cohen's $d = 0.8$; late: $t(7) = 2.8$, p = 0.03, Cohen's $d = 1.0$).

We used linear mixed-effects modeling (LME) to determine whether response component amplitudes differed for peak versus trough stimulation across subjects. We constructed an LME model for each response component with a fixed effect of phase and random effects on slope and intercept of both subject and electrode nested within subject [i.e., response amplitude ~ phase + (phase|subject/electrode)]. We then performed likelihood ratio tests to assess whether this model fit was improved

by the inclusion of the fixed effect of phase. That is, after accounting for participant variance, we tested whether stimulation phase improved the prediction of response amplitude. Including the phase effect improved the model fit for the early component, with a marginal effect in the late component (one-sided likelihood ratio test for improvement, early: LRStat = 3.8, p = 0.05; late: LRStat = 3.3, p = 0.06). We then repeated this process using amplitude values from the isolated evoked response (i.e., following correction for non-evoked oscillatory activity, as in *Figure 3*). The effect of phase was reduced in these models (early: LRStat = 2.8, p = 0.09; late: LRStat = 0.9, p = 0.3), in accordance with the observed reduction in peak versus trough differences at the electrode level following correction for non-evoked oscillatory activity.

## Discussion

We investigated whether the human hippocampus varies in receptivity to external stimulation along with the local theta oscillation. Lateral temporal stimulation consistently evoked a hippocampal response with distinct early and late negative components. These component amplitudes were found to vary continuously with theta phase at stimulation onset. The continuous relationship between phase at stimulation onset and EP amplitude showed temporal specificity, with a notable increase during the early component. We additionally found that stimulation at theta peak versus theta trough yielded maximal differences in evoked response amplitude. This effect was most pronounced during the early negative component (27–113 ms after stimulation onset), where it persisted even after corrections for non-evoked oscillatory activity which emerged from the phase-sorting procedure. These findings suggest that human hippocampal connectivity to network afferents varies across the local theta oscillation.

As data in this study were collected opportunistically from participants undergoing clinically necessary invasive monitoring, hippocampal electrodes were implanted at variable laminar depths that could not be known due to the relative imprecision of CT/MRI. Nonetheless, we observed a consistent effect of phase angle on the recorded hippocampal response. The relatively small phase offset we observed between hippocampal sites from a given participant (see *Appendix 1—figure 4*) supports that the recorded oscillation was depth indifferent, perhaps owing to the large size and interlaminar placement of the macroelectrodes. Further studies would be necessary to determine subfield and layer-specific phase angles conferring maximum entorhinal input to the human hippocampus.

The observed effect of peak versus trough is likely specific to the synaptic distance and conduction delay between the stimulation site and hippocampus. While stimulating electrodes were all located in lateral temporal cortex, their gyral locations varied according to clinical constraints. Accordingly, while phase offsets between lateral temporal cortex and hippocampus were non-uniform (see *Appendix 1—figure 4*), there were pronounced differences in the offset angle between participants. It is possible that differences in transmission latencies contributed to the observed variance in peak–trough effects across participants. We hypothesize that stimulation targeting a different site in the hippocampal network would produce a similarly theta-periodic hippocampal response, albeit likely maximized at a different stimulation phase angle (i.e., not necessarily at the observed hippocampal theta peak). Further, it is possible that the relationship of theta phase to the evoked response amplitude varies with theta power, although we were methodologically unable to assess theta power on a per-trial basis. Our exploratory analyses of theta power during an offline period did not find any relationship between power and the phase–amplitude relationship (see Appendix 1: Offline theta bout incidence does not impact periodicity of the evoked response), although quantifying power during a period other than stimulation is not ideal. Future research could address this issue, potentially by cueing stimulation based on an online assessment of theta power.

One limitation is that this study was performed in individuals with refractory epilepsy. Temporal lobe epilepsy is associated with episodic memory impairments (*Mayeux et al., 1980*; *Helmstaedter and Kockelmann, 2006*) thought to be caused by structural abnormalities as well as interictal epileptiform activity in the hippocampus and hippocampal cortical network (*Helmstaedter and Kockelmann, 2006*; *Gelinas et al., 2016*). Nonetheless, iEEG recordings from individuals with temporal lobe epilepsy have previously been used to study mechanisms for hippocampal function (*Lega et al., 2012*; *Long et al., 2014*; *Fell et al., 2011*; *Wixted et al., 2014*). To enhance the study's generalizability, we rejected trials with recorded epileptiform activity and ensured during data collection that

the stimulation protocol did not elicit afterdischarges or spiking. It is nonetheless unclear whether epilepsy-related changes to hippocampal structure and network connectivity influenced our findings.

Phase dependence of the response to stimulation has, however, been demonstrated in non-epilepsy model organisms. Previous studies have reported phase-dependent responses to external stimulation across diverse neocortical areas. Direct electrical stimulation of sensory cortices has been found to differentially induce long-term potentiation or depression depending on LFP phase e.g., with beta- and gamma-dependence in rodent visual cortex (*Wespatat et al., 2004*) and beta-dependence in primate sensorimotor cortex (*Zanos et al., 2018*). In humans, local oscillatory phase has been found to relate to the amplitude of the cortical response evoked by transcranial magnetic stimulation (*Kundu et al., 2014*). These findings support that oscillatory phase relates generally to local excitability. Thus, one part of our analysis strategy was to assess whether any observed phase dependence was specific to (or specifically enhanced in) hippocampus. We investigated the phase dependence of stimulation response in two control regions: amygdala and orbitofrontal cortex. Like hippocampus, both regions have anatomical (*Shi and Cassell, 1997*; *Iwai et al., 1987*; *Morecraft et al., 1992*) and functional (*Roy et al., 2009*; *Du et al., 2020*) connectivity with the lateral temporal stimulation site. But while amygdala is physically adjacent to hippocampus and densely connected with hippocampus and entorhinal cortex (*Saunders et al., 1988*; *Pikkarainen et al., 1999*; see *Chrobak et al., 2000*) – and might therefore be expected to show hippocampus-like phase dependence of input receptivity – orbitofrontal cortex is more distant both in space and connectivity. Indeed, we observed significant theta-phase dependence in the amygdala EP but no such effect in orbitofrontal cortex. As was the case in hippocampus, the amygdala showed greater response amplitude when stimulation was applied at theta peak relative to trough. The amygdala also showed greater response amplitude to stimulation at theta rising phase relative to falling phase. Unlike in hippocampus, this effect was present exclusively in the late component. The presence of early components in the amygdala and orbitofrontal EPs supports connectivity between these control regions and the lateral temporal stimulation site. However, we note that the stimulation site was chosen on the basis of its functional connectivity with hippocampus as measured via the stimulation-evoked potential, and not based on connectivity with these other brain areas. Performing network-targeted stimulation for each control region could provide stronger evidence for the effect's selectivity to hippocampus.

This study used direct electrical stimulation as a proxy for endogenous network signaling. Follow-up studies are required to assess whether and how these changes in human hippocampal connectivity due to theta phase relate to memory processing. Nonetheless, by demonstrating phase dependence of input receptivity in the human hippocampus, this study suggests a homology with the phase dependence previously characterized in rodent models in relation to memory encoding and retrieval.

## Materials and methods

### Electrode localization

sEEG electrodes were localized using MRIcron (v1.0.20190902; *Rorden and Brett, 2000*) and the Statistical Parametric Mapping package (SPM12; *Penny et al., 2011*). Pre-implant T1-weighted structural MRI and post-implant computed tomography (CT) were acquired as part of clinical care. For each subject, we performed tissue-type segmentation on the MRI (with default SPM12 tissue probability maps and warping parameters; see *Ashburner and Friston, 2005*; *Mechelli et al., 2005*) then normalized the MRI to MNI space (ICBM Average Brain template MNI152; *Mazziotta et al., 1995*). We applied this same transformation to the CT, which had been co-registered to the MRI by normalized mutual information. We then localized electrodes within MNI space by visual inspection of the CT. The anatomical location of each electrode was confirmed by atlas-guided inspection of the MRI (Allen Human Brain Atlas; *Ding et al., 2016*). We were unable to obtain imaging data for one subject and therefore relied on the electrode localization provided by the clinical team (comprising surrounding tissue type and anatomical structure for each electrode).

### sEEG recording and stimulation

sEEG depth electrodes (~1 mm diameter, ~2 mm contact length, 5–10 mm contact spacing; AD-Tech, Oak Creek, WI) were implanted prior to study participation according to clinical need. Recordings were acquired using a Neuralynx ATLAS system with a scalp electrode reference and ground. Data

were recorded at a resolution of 0.15 µV (5000 µV input range) and a sampling rate of 20 or 32 kHz. Digital bandpass filters (FIR) from 0.1 to 5000 Hz were applied at the time of recording. Data were re-referenced offline to the common average of ipsilateral depth electrodes (*Zhang and Jacobs, 2015*; *van der Meij et al., 2012*) and downsampled to 1 kHz. Data were epoched about stimulation pulses and baseline corrected (epoch: −750 to 500 ms, baseline: −750 to −2 ms). To prune excessively noisy or artifactual data, epochs were excluded according to their signal range (excluded if >800 µV) and kurtosis (excluded if >2 SD over channelwise mean kurtosis. Mean epochs pruned per channel ± SD, hippocampus: $n$ = 208 ± 329 epochs; amygdala: $n$ = 275 ± 333 epochs; orbitofrontal cortex: $n$ = 250 ± 347 epochs. Mean epochs included in analyses per channel ± SD, hippocampus: $n$ = 1194 ± 599 epochs; amygdala: $n$ = 1341 ± 348 epochs; orbitofrontal cortex: $n$ = 1479 ± 263 epochs). Channels were excluded from analyses if <200 epochs remained following pruning. $n$ = 3 hippocampal channels, $n$ = 3 amygdala channels, and $n$ = 5 orbitofrontal channels were excluded from analyses on this basis.

Electrical stimuli were generated with a Grass Instruments S88 stimulator in conjunction with CCU1 constant current units and SIU5 stimulus isolators. Stimulation was delivered across two adjacent lateral temporal sEEG electrodes. The electrical stimulus comprised a constant-current, symmetric-biphasic square wave with 5 mA intensity and 0.6 ms total duration. Stimulation polarity was reversed across the two electrodes such that stimulation on the lateral electrode was anodic-leading and stimulation on the medial electrode was cathodic-leading. For simplicity we refer to each electrical stimulus as a 'single pulse'.

Stimulating electrodes used for the experiment were selected during a preliminary stimulation session to identify electrodes with hippocampal functional connectivity (i.e., for which stimulation would evoke downstream hippocampal EPs) and for which stimulation would not be clinically problematic. Potential stimulating electrode pairs were identified in lateral temporal cortex and adjacent white matter based on well-characterized structural and functional connectivity of these regions with ipsilateral entorhinal cortex and hippocampus (*Insausti et al., 1987*; *Catani and Thiebaut de Schotten, 2008*; *Zhong and Rockland, 2004*). Of these, we excluded electrodes where stimulation provoked seizure or afterdischarges during clinical testing. To evaluate functional connectivity with hippocampus, trains of stimulation were delivered to each potential electrode pair (0.5 Hz; 30 pulses per pair). Mean evoked potentials (EPs) for each hippocampal electrode were visualized in real-time and manually inspected. The lateral temporal electrode pair for which stimulation elicited the largest mean EP for hippocampal electrodes was selected for the experimental protocol. A single pair of stimulating electrodes was selected for each participant ($n$ = 16 total stimulating electrodes).

Participants remained in bed throughout the preliminary stimulation session and experimental session. They were not instructed to perform any task and were free to rest or otherwise occupy themselves. Study protocols were approved by the Northwestern University Institutional Review Board. All subjects provided written informed consent prior to participation.

## Theta-phase estimation

We estimated hippocampal theta (3–8 Hz) phase at the time of stimulation onset for each trial. This was done separately for each electrode because theta-phase shifts across space in the hippocampus. In depth penetrations of rodent CA1, theta phase is stable through the strata oriens and pyramidale then undergoes a gradual phase shift, resulting in a 180° phase difference at the fissure compared to dorsal layers (*Brankack et al., 1993*; *Bragin et al., 1995*). Rodent hippocampal theta phase has been found to fully reverse between the longitudinal poles (*Patel et al., 2012*) i.e., the 'traveling wave' model (*Patel et al., 2012*; *Lubenov and Siapas, 2009*). Monotonic phase shift across the long axis has also been demonstrated to occur within individual CA subfields (*Lubenov and Siapas, 2009*). In LFPs recorded along the human hippocampus via implanted depth electrodes, theta phase has been found to shift monotonically across the hippocampal long axis (*Zhang and Jacobs, 2015*). Studies of hippocampal theta in rodent models often estimate phase at the stratum lacunosum-moleculare of CA1, near the hippocampal fissure. Besides the benefit of fissural theta's especially high amplitude (*Brankack et al., 1993*), this approach provides a unitary measure of hippocampal theta phase. In humans, it is not feasible to control for these spatial phase shifts by recording uniformly at any specific site in hippocampus. This is because electrodes are placed according to clinical need, and their locations vary across subjects in both laminar depth and location on the long axis.

LFPs recorded from human depth electrodes generally reflect a sum of phase-asynchronous laminar inputs (*West and Gundersen, 1990*) weighted by distance from the contributing layer to the electrode. As the phase shift is stable across layers and septotemporal distance (*Lubenov and Siapas, 2009*), phase at a given electrode has a consistent offset to other hippocampal sites (*Zhang and Jacobs, 2015*). Thus, to account for this offset, we estimated phase independently for each electrode. First, trial epochs were truncated at +50 ms (i.e., 50 ms following stimulation onset) to avoid contamination of the phase estimate by the stimulation EP. After applying a zero-phase bandpass-filter (3–8 Hz second-order Butterworth IIR), we estimated phase angle at the time of stimulation onset using the Hilbert transform.

We assessed the accuracy of this approach using stimulation-free pseudotrials. For each hippocampal electrode, we first performed zero-phase bandpass filtering (3–8 Hz, second-order Butterworth IIR) across a continuous stimulation-free period (from the same recording as used in the main analysis). We then applied the Hilbert transform to obtain ground-truth phase angles for each timepoint. We created stimulation-free trials by pseudorandomly selecting trial-length epochs during this stimulation-free period. We observed the ground-truth phase values at the timepoints corresponding to each trial's mock stimulation onset. Epochs from the first and last 5 s of the recording sessions were excluded to reduce filter edge artifact contamination.

Next, we created an ostensibly phase-balanced model of stimulation for each channel by binning stimulation trials at 90° intervals (centered at 0°, 90°, 180°, and 270°) and computing the grand average across bins. We added this model stimulation to each stimulation-free trial, yielding pseudotrials. We estimated pseudotrial phase angle at $t = 0$ using the approach described for stimulation trials and compared the estimates to the ground-truth phase angles.

## Analysis of narrowband theta activity

To assess whether theta-frequency activity was present in hippocampus and control regions, we characterized narrowband activity across the power spectrum during a continuous stimulation-free period (from the same recording as used in the main analysis; all rest periods had duration >90 s). We estimated oscillatory power at 50 logarithmically spaced frequency intervals from 1 to 50 Hz using the fast Fourier transform. To identify frequencies where reliable oscillations were present, we first estimated the background $1/f$ power spectrum using a robust linear fit to the log–log scaled power spectrum (*Lega et al., 2012*). We subtracted this background from the power spectrum and identified positive local maxima in the resultant curves, following four-frequency boxcar smoothing to eliminate noisy peaks (*Lega et al., 2012*). We performed this analysis for individual electrodes as well as for the mean power spectrum across all analyzed electrodes within each ROI. For each identified peak frequency, we assessed whether its power consistently exceeded the background $1/f$ spectrum by performing one-sample $t$-tests of the corrected spectrum power versus 0 across electrodes.

## Quantification of hippocampal EPs

As a measure of the hippocampal response to stimulation, we quantified the trialwise amplitudes of early and late components in the stimulation EP. We first estimated component timecourses for each electrode. To avoid phase-dependent differences in component shape or timecourse from biasing this estimate (i.e., in the case of non-uniform stimulation phase distributions), we computed a phase-balanced EP for each electrode by binning stimulation trials according to theta phase at stimulation onset (at 90° intervals, centered at 0°, 90°, 180°, and 270°) and computing the mean across bins. We then observed the grand average phase-balanced trial across electrodes. We quantified component timecourses by searching for the first two negative minima following stimulation artifact (on a search window of +20 to +500 ms after stimulation; see e.g., *Kubota et al., 2013*) on the grand average trial across electrodes. Component edges were estimated as the nearest inflection points within 150 ms of the local maximum. A minimum interval of 50 ms was required between peaks. For each trial, we computed the average signal amplitude across each component timecourse. This method was selected rather than peak estimation (as in e.g. *Matsumoto et al., 2004*) in order to produce a more noise-indifferent estimate for single trials.

## Circular–linear analysis of theta phase and hippocampal response amplitude

We performed circular–linear analyses to determine whether the hippocampal response varied continuously with theta phase at stimulation onset. For each electrode, we found the circular–linear correlation coefficient between phase at stimulation onset and component amplitude (*Berens, 2009*). We z-scored these values via permutation testing (*n* = 500), wherein each electrode's trial phase values and component amplitudes were repeatedly randomly paired. The circular–linear correlations were evaluated using a one-sample *t*-test comparing the electrode z-scores against zero. To investigate the timecourse of the continuous phase–amplitude relationship, we performed a follow-up timepoint analysis. For each electrode, we computed the circular–linear correlation coefficient between phase at stimulation onset and EP amplitude at every timepoint. We identified peaks in the phase–amplitude relationship by searching for the first two maxima following stimulation artifact (on a search window of +20 to +500 ms after stimulation). This method was modified from our procedure to identify components in the EP amplitude timecourse (see Quantification of hippocampal EPs).

## Comparison of hippocampal EPs by binned theta phase

To test how specific phase angles were related to hippocampal responsiveness to stimulation, we analyzed trials according to theta phase at stimulation onset. We estimated local broadband theta phase at stimulation onset for each trial and binned trials to 90° intervals, centered on peak, trough, rising, and falling phase angles. By taking the means within each bin, we obtained average peak, trough, rising and falling stimulation trials for each electrode. We then compared component amplitudes across peak versus trough and rising versus falling trials using paired *t*-tests.

## Comparison of stimulation trials to phase-matched stimulation-free trials

Direct comparison of component amplitudes according to phase at stimulation onset is complicated because oscillatory phase necessarily predicts future amplitude, regardless of any phase-dependent differences in the effects of stimulation on EPs. We therefore used stimulation-free trials to account for the ongoing theta oscillation. Stimulation-free trials were captured for each electrode at 100-ms intervals across stimulation-free periods at the beginning and end of the recordings. Stimulation-free trials were recorded and preprocessed using the same approach as stimulation trials (for trial pruning by kurtosis, kurtosis scores were compared only to other stimulation-free trials from the same channel). Phase was estimated for each trial at *t* = 0 using the same methods as for stimulation trials (i.e., truncating at +50 ms after stimulation onset, filtering, and applying the Hilbert transform). In order to more closely match the non-evoked activity present in the stimulation trials, we randomly resampled stimulation-free trials using the phase angle distributions of the stimulation trials as sampling weights.

## Timepoint analysis of EP amplitude following stimulation at theta peak versus trough

As a follow-up to our finding of peak versus trough effects on component amplitudes, we investigated whether phase dependence was temporally restricted to components. For each electrode, we computed the difference between average peak and trough trials at each timepoint. To assess the contribution of the non-evoked signal over time, we repeated this procedure on stimulation-free trials. To measure phase dependence related to the evoked signal, we compared peak–trough amplitude differences across stimulation and stimulation-free trials.

## Oscillatory synchronization between the stimulation site and hippocampus

Although we estimated hippocampal theta-phase angle at the time of stimulation onset to avoid bias from the evoked response, this timepoint is not the most relevant to hippocampal receptivity to external input. As stimulation was applied to lateral temporal network afferents and conveyed via polysynaptic signaling, there was likely some latency between stimulation onset and the relevant transmission to hippocampus (i.e., the timepoint when entorhinal input receptivity would be relevant).

Oscillatory synchronization (i.e., phase coupling) is a known mechanism that supports interregional communication (see *Fries, 2005*; *Fell and Axmacher, 2011*). We therefore estimated this latency by observing theta-phase locking and phase offset between the stimulation site and hippocampus. First, we estimated 3–8 Hz theta-phase angle for each electrode across a continuous, stimulation-free period in the recording. For each timepoint, we then obtained the angular distance between each hippocampal electrode and its corresponding lateral temporal electrode that was used for stimulation. We thereby computed the phase-locking value (PLV; mean resultant vector length of lateral temporal–hippocampal angular distance on the unit circle) and mean phase offset (the circular mean of lateral temporal–hippocampal angular distances) for each hippocampal electrode.

We applied the Rayleigh test to the mean phase offsets to assess whether the phase lag distribution was uniform. We used permutation testing to determine whether the observed phase locking was greater than expected by chance. To achieve this, we broke the continuous hippocampal and lateral temporal phase estimates into 500-ms epochs. We then obtained the mean PLV across all epochs. To z-score these PLVs, we used permutation testing ($n = 500$), wherein hippocampal and lateral temporal epochs were repeatedly randomly paired. We performed a one-sample t-test comparing z-scores against zero to assess phase locking.

## Acknowledgements

We thank the participants and their families for generously volunteering their time to this research. We thank Ania Holubecki and the staff of the Northwestern Memorial Hospital Epilepsy Monitoring Unit for coordinating data collection. We thank Dr. Christina Zelano, Dr. Matt Oh, and Dr. John Disterhoft for their helpful comments and feedback and Dr. Jim Baker for his guidance in hardware setup and troubleshooting. The content is solely the responsibility of the authors and does not necessarily represent the official view of the National Institutes of Health.

## Additional information

### Funding

| Funder | Grant reference number | Author |
| --- | --- | --- |
| National Institute of Neurological Disorders and Stroke | R01NS113804 | Joel L Voss |
| National Institute of Mental Health | F31MH125577 | Sarah M Lurie |

The funders had no role in study design, data collection, and interpretation, or the decision to submit the work for publication.

### Author contributions

Sarah M Lurie, Conceptualization, Data curation, Software, Formal analysis, Investigation, Visualization, Methodology, Writing - original draft; James E Kragel, Conceptualization, Data curation, Investigation, Writing - review and editing; Stephan U Schuele, Supervision, Investigation, Methodology, Writing - review and editing; Joel L Voss, Conceptualization, Supervision, Methodology, Writing - review and editing

### Author ORCIDs

Sarah M Lurie http://orcid.org/0000-0003-2986-688X

### Ethics

All subjects provided written informed consent prior to participation. Study protocols were approved by the Northwestern University Institutional Review Board (STU00210599).

### Decision letter and Author response

Decision letter https://doi.org/10.7554/eLife.78395.sa1
Author response https://doi.org/10.7554/eLife.78395.sa2

## Additional files

### Supplementary files
• Transparent reporting form

### Data availability
All iEEG data and custom analysis scripts have been made publicly available on Zenodo (https://doi.org/10.5281/zenodo.6342237).

The following dataset was generated:

| Author(s) | Year | Dataset title | Dataset URL | Database and Identifier |
|---|---|---|---|---|
| Lurie SM, Kragel JE, Schuele SU, Voss JL | 2022 | Data and scripts pertaining to "Human hippocampal responses to network intracranial stimulation vary with theta phase" | https://doi.org/10.5281/zenodo.6342237 | Zenodo, 10.5281/zenodo.6342237 |

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

## Appendix 1

### Phase angle estimates were consistent with ground-truth phase values in analysis of stimulation-free pseudotrials

The method we used to estimate hippocampal theta phase at the time of stimulation involved truncating the hippocampal recordings shortly after each stimulation pulse (see Materials and methods: Theta-phase estimation). We assessed whether this approach yielded an accurate estimate of phase (i.e., one based on the ongoing oscillatory activity and without contamination by the evoked response or filter artifact) using 'pseudotrials', epochs made from data collected during a continuous, stimulation-free period with an added model stimulation pulse and EP. We estimated the phase of these pseudotrials at mock stimulation onset (i.e., $t = 0$) using the same approach as for stimulation trials. As pseudotrials were created from continuous, stimulation-free data, we were also able to calculate 'ground-truth' phase values (i.e., obtained in the absence of stimulation artifact or EP and without truncating the recording). We then measured the trialwise differences between ground-truth and estimated phase angles.

Across hippocampal recording electrodes, the mean difference between ground-truth and estimated phase was −11.2° (mean distance ± SD: −11.2 ± 4.6°). This difference was highly concentrated (Rayleigh test: $z(22) = 22.9$, $p < 0.001$), indicating consistency of the phase angle estimate performance across electrodes (*Appendix 1—figure 1a*). Although estimated phase angles were significantly more concentrated than ground-truth phase angles (two-tailed $t$-test on mean resultant vector lengths, $t(22) = 7.4$, $p < 0.001$, Cohen's $d = 2.1$), the distribution of mean estimated phase angles was uniform (Hodges–Ajne test: $m(22) = 9$, $p = 0.9$). These findings demonstrate that the estimation approach introduced a small and consistent phase angle bias.

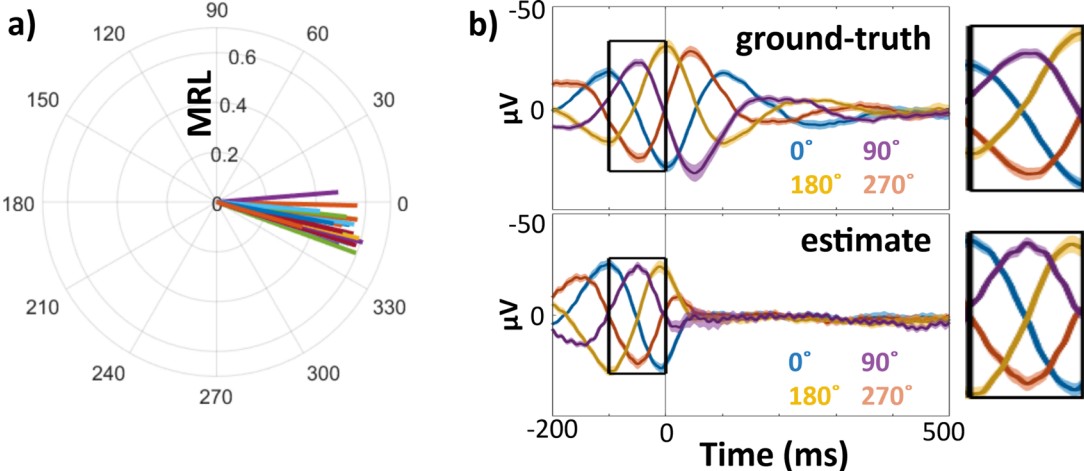

**Appendix 1—figure 1.** Phase estimation method yielded consistent, low-magnitude offset to ground-truth phase in pseudotrial analysis. Comparison of ground-truth and estimated phase angles from pseudotrials (stimulation-free resting data with added model stimulation artifact and EP). Validation was performed across all hippocampal channels included in the main EP analysis ($n = 23$). (**a**) Mean phase angle and mean resultant length (MRL) of trialwise distance between ground-truth and estimated phases. Each line represents one hippocampal channel. (**b**) Left: Mean stimulation-free validation trials binned according to ground-truth (top) and estimated (bottom) phase at mock stimulation onset. Line color indicates bin center (0°, 90°, 180°, and 270°). Right: enlarged panels from −100 to 0 ms, showing similarity of theta phase at $t = 0$ across ground-truth (top) and estimated (bottom) bins.

We also observed differences in the intertrial theta coherence across trials binned by ground-truth versus estimated phase (*Appendix 1—figure 1b*). Because the phase estimation approach involves truncating the epoch at +50 ms following stimulation onset, signal after this point does not contribute to the phase estimate. Theta coherence therefore decreases asymmetrically about $t = 0$, with a more rapid dropoff after $t = 0$ than before it. In contrast, trials binned according to ground-truth phase show symmetrical declines in coherence before and after $t = 0$.

To assess whether this bias impacted our analyses, we re-performed our binning analyses after accounting for each electrode's estimated bias (*Appendix 1—figure 1b*). In other words, we adjusted the phase angle label for each trial according to the electrode-specific bias. After this correction, we

observed the same pattern of phase dependence as in the original analysis, with enhancement of the EP by stimulation at peak versus trough in both early and late components (early: $t(22) = -2.5$, p = 0.02, Cohen's $d = -0.2$; late: $t(22) = -2.7$, p = 0.01, Cohen's $d = -0.09$) and no effect of falling versus rising phase stimulation (early: $t(22) = -0.22$, p = 0.8, Cohen's $d = -0.02$; late: $t(22) = 0.15$, p = 0.9, Cohen's $d = 0.007$). Rebinning did not impact the results of the continuous, circular–linear analyses.

## Order of stimulation pulses had no effect on phase distribution

One possible alternative explanation for the observed relationship between phase and EP amplitude was that over the course of the experimental session, stimulation induced changes in both the distribution of phase angles at stimulation onset and in the amplitude of the evoked response. We therefore performed control analyses to assess whether the theta-phase distribution of stimulation pulses and their associated hippocampal evoked responses changed according to order in the experimental session.

For each hippocampal electrode, we divided stimulation trials into quartiles according to order of occurrence in the stimulation session. Within each quartile, we estimated the uniformity of stimulation phase distributions via Rao's spacing test statistic $u$ (*Appendix 1—figure 2a*). Across electrodes, there was no change in the uniformity of stimulation phase distributions by order of occurrence (one-way repeated measures analysis of variance (ANOVA), effect of quartiles: $f(3) = 2.02$, p = 0.1; electrode:quartile interaction: $f(3) = 1.29$, p = 0.07. Mean Rao's $u$ ± SD, quartile 1: $m = 148.0 ± 14.5$; quartile 2: $m = 149.1 ± 10.4$; quartile 3: $m = 148.6 ± 12.8$; quartile 4: $m = 148.3 ± 13.3$). Further, we observed no difference in mean phase angles across quartiles (*Appendix 1—figure 2b*. Watson–Williams test: $f(3) = 0.02$, p > 0.99). We also assessed mean early and late 1 and N2 component amplitude by quartile (component amplitudes calculated as in Materials and methods: Quantification of hippocampal EPs). We observed an effect of quartile specifically on the late component amplitude (one-way repeated measures ANOVA, effect of quartiles: $f(3) = 4.3$, p = 0.008; electrode:quartile interaction: $f(3) = 3.1$, p = 0.03). No such effect was observed for the early component (one-way repeated measures ANOVA, effect of quartiles: $f(3) = 0.61$, p = 0.6; electrode:quartile interaction: $f(3) = 0.10$, p > 0.9). The absence of concomitant changes in stimulation phase distribution and evoked amplitude across quartiles supports that the observed phase–amplitude relationship was not driven by ordering in the experimental session.

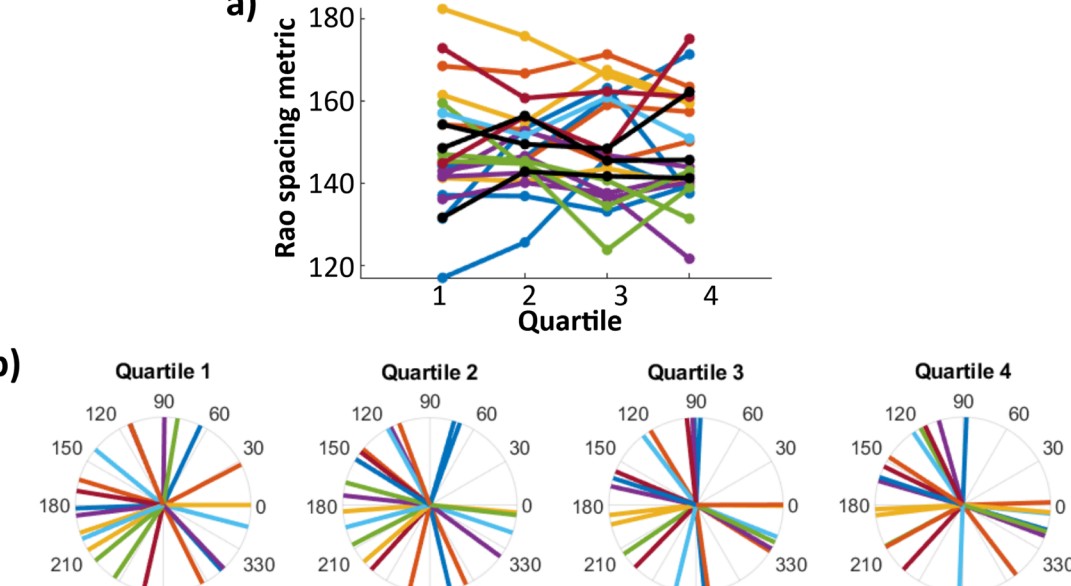

**Appendix 1—figure 2.** Theta-phase distribution of stimulation pulses did not change according to order in the experimental session. Color indicates participant of origin. (**a**) Rao's spacing test statistic $u$ for stimulation phase angles in first, second, third, and fourth quartiles of the experimental session. Each dot is one hippocampal electrode ($n = 23$). (**B**) Mean stimulation phase angle for each hippocampal electrode across quartiles. No significant differences are found in either distribution uniformity or directionality according to order in the experimental session.

## Hippocampal peak versus trough EP amplitude differences were temporally localized to components

To complement the analyses of component amplitudes, we also performed an exploratory analysis of peak versus trough amplitude differences across all timepoints in the peri-stimulus trial period. As in the analyses above, we compared amplitudes across stimulation and stimulation-free trials in order to hone in on the stimulation-evoked response controlling for expected amplitude differences due to phase in the absence of stimulation. As expected, both stimulation and stimulation-free trials exhibited strong peak versus trough differences before t = 0 (*Appendix 1—figure 3a and b*). In the stimulation-free trials, the peak-trough difference drops off asymmetrically about t=0 (*Appendix 1—figure 3b*), likely related to the phase estimation method (which was used for both stimulation and stimulation-free trials and involved truncating each trial at +50 ms; see *Appendix 1—figure 1b*). Isolating the evoked response revealed that peak stimulation selectively enhanced signal negativity during the identified EP components, with consistent peak versus trough differences throughout the early component in particular (*Appendix 1—figure 3c*). Notably, following stimulation, there were no peak versus trough differences until after approximately +40 ms, indicating that effects of phase on the subsequent amplitude of the EP components were not due to lingering differences in the ongoing oscillation irrespective of stimulation. These findings suggest that the effects of stimulation phase were temporally selective to the EP components.

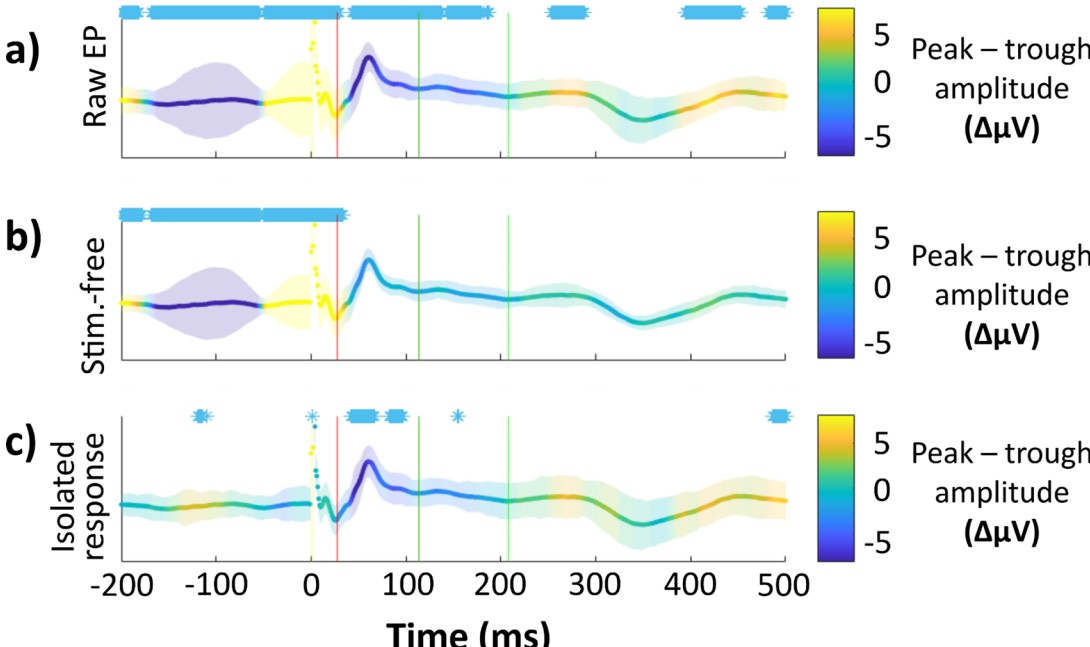

**Appendix 1—figure 3.** Effect of peak versus trough stimulation on hippocampal isolated response amplitude is temporally specific to components. The difference between peak and trough amplitude at each timepoint is colorized and plotted on top of the grand average hippocampal EP. Purple indicates greater negativity for peak stimulation trials, yellow indicates greater negativity for trough stimulation. Error bars indicate ±1 standard error of the mean (SEM) of the peak − trough amplitude difference (a.u.). Timepoints where p < 0.05 (uncorrected, two-tailed) for non-zero amplitude difference between peak and trough are marked (*) for visualization. Component boundaries are marked with vertical lines (early: red, late: green). Plotted for: (**a**) Raw EPs (i.e., stimulation trials). (**b**) Stimulation-free trials only. (**c**) Isolated evoked response (i.e., stimulation trials minus stimulation-free trials).

## Offline theta bout incidence does not impact periodicity of the evoked response

In humans and non-human primates, hippocampal theta oscillations occur in intermittent, high-power bouts, typically lasting only a few theta cycles (*Goyal et al., 2020*). Bout incidence in the hippocampus and broader hippocampal network has been associated with spatial navigation (*Kahana et al., 1999*) and performance of non-spatial memory tasks (*Raghavachari et al., 2001*). Given the relevance of theta bouts to hippocampal function, we investigated whether the observed

theta-phase dependence of response amplitude was stronger in electrodes with greater bout incidence. As estimates of theta power during the stimulation protocol would be contaminated by the evoked response, we analyzed theta bout incidence during stimulation-free rest just prior to the experimental session. We used this measure as a proxy for the proportion of stimulation trials we expected to occur during periods of high theta power.

For each hippocampal electrode, we performed bout detection in the 3–8 Hz range during continuous periods of stimulation-free rest using the BOSC toolbox (see *Whitten et al., 2011*; *Hughes et al., 2012*). First, we computed power across the stimulation-free period using the Morlet wavelet transform ($w_0$ = 6; 50 frequency samples spaced logarithmically from 1 to 60 Hz). To isolate significant oscillations, we fit a 1/$f$ background power spectrum for each electrode by performing robust linear regression on the time-averaged power spectrum in log–log space. Then, for each sampled frequency, we identified timepoints where power exceeded the 95th percentile of the estimated $\chi^2$ probability distribution of the background spectrum for a minimum duration of three cycles. We thereby obtained a theta bout rate for each hippocampal electrode corresponding to the proportion of the recording where high-power oscillations were present in any of the 3–8 Hz frequency samples.

Across electrodes, detected bout incidence ranged from 27% to 73% of the recording (mean incidence ± SD = 48 ± 13%). We quantified response amplitude theta periodicity for each electrode as the *z*-scored circular–linear correlation between hippocampal theta phase at stimulation onset and component amplitude. Bout rate was not associated with response theta periodicity in either the early (linear correlation: $r$ = 0.27, p = 0.2) or late ($r$ = 0.0030, p > 0.9) components. While we were methodologically unable to quantify the effect of bouts on individual trials, this finding suggests that resting bout incidence did not impact the phase dependence of the hippocampal evoked response.

## Accounting for estimated phase latency between the stimulation site and hippocampus does not produce consistent phase angles conferring maximal and minimal EP amplitude

There is necessarily a conduction delay between the lateral temporal stimulation site and its receipt in hippocampus. As functionally connected regions are frequently phase-synchronized (see *Fell and Axmacher, 2011*), we hypothesized that this delay would translate to a consistent theta-phase lag between the stimulating electrodes and the hippocampus. This would produce a consistent angle offset between hippocampal theta phase at the time of stimulation (i.e., at $t$ = 0, the timepoint which we used to characterize trial phase in the previous analyses) and the stimulation's arrival at hippocampus.at the time of the relevant entorhinal–hippocampal transmission.

Permutation testing (see Materials and methods: Oscillatory synchronization between the stimulation site and hippocampus) revealed significant phase locking between hippocampus and the stimulation site (mean *z*-score ± SD: z = 3.1 ± 4.6; *t*-test of *z*-scores against 0: $t(22)$ = 3.2, p = 0.004, Cohen's $d$=0.7. Mean PLV ± SD: 0.18 ± 0.1), indicating that individual hippocampal electrodes had consistent phase lags to the stimulation site. However, across electrodes, the distribution of mean phase offsets was not significantly non-uniform (*Appendix 1—figure 4a*. Rao's spacing test: $u(22)$ = 144.9, p = 0.5). We noted a bimodal distribution of high-PLV phase latencies across electrodes, with clusters centered approximately 180° apart. Consequently, across electrodes with consistent phase lags, stimulation delivered during the hippocampal peak may have arrived at to hippocampus at different phase angles across electrodes.

To assess whether the variability in transmission delay across electrodes impacted the observed relationship between theta phase and evoked response amplitude, we reanalyzed the relationship between component amplitude and specific theta-phase angles (see *Figures 3 and 4*) after accounting for the observed latency between the hippocampal electrode and the stimulation site. For example, for a hippocampal electrode with an observed phase lag of 180°, stimulation delivered at the hippocampal theta peak would arrive on average at the subsequent hippocampal theta trough. However, after rebinning trials according to latency, we found no difference across either peak versus trough (early: $t(22)$ = 1.7, p = 0.1, Cohen's $d$ = 0.2; late: $t(22)$ = 1.4; p = 0.2, Cohen's $d$ = 0.08) or rising versus falling (early: $t(22)$ = 1.1, p = 0.3, Cohen's $d$ = 0.1; late: $t(22)$ = 0.30, p = 0.8, Cohen's $d$ = 0.01) component amplitudes. The rebinning procedure had no effect on the unbinned, circular–linear analyses.

We observed clustering of phase latencies near 0° and 180° (*Appendix 1—figure 4a*), a possible indicator of volume conduction between the two sites. To investigate, we reanalyzed phase locking between hippocampus and the stimulation site after adopting a bipolar referencing scheme wherein data were re-referenced to the adjacent, lateral contact on the depth electrode (rather than the average of ipsilateral depth electrodes, as in all previous analyses). After re-referencing, the significant phase locking between hippocampus and the stimulation site persisted (*Appendix 1—figure 4b*. Mean z-score ± SD: z = 2.3 ± 3.8; *t*-test of z-scores against 0: *t*(22) = 2.9, p = 0.009, Cohen's *d* = 0.6. Mean PLV ± SD: 0.15 ± 0.1). The qualitative clustering of phase latencies near 0° and 180° were also present in the re-referenced analysis.

After rebinning trials according to the latency estimated under bipolar referencing conditions, we found a significant difference across rising versus falling late component amplitudes (early: *t*(22) = −0.57, p = 0.6, Cohen's *d* = −0.06; late: *t*(22) = −2.35, p = 0.03, Cohen's *d* = −0.1), which persisted following correction for the non-evoked oscillation (late: *t*(22) = −2.50, p = 0.02, Cohen's *d* = −0.1. Abolition of the non-evoked oscillation was performed using the same procedure as in *Figure 3*; see Materials and methods: Comparison of stimulation trials to phase-matched stimulation-free trials). No differences were found across peak versus trough trials for either the raw EP (early: *t*(22) = −0.18, p = 0.9, Cohen's *d* = −0.01; late: *t*(22) = 0.04, p > 0.9, Cohen's *d* = 0.002) or the isolated response (early: *t*(22) = −0.31, p = 0.8, *d* = −0.04; late: *t*(22) = −0.33, p = 0.7, Cohen's *d* = −0.02).

Qualitatively, phase offsets from within the same participant were highly concentrated, indicating low theta-phase shift across hippocampal electrodes despite variable laminar depths and septotemporal placement within individual subjects. While previous studies have reported relatively small phase angle shifts across the human hippocampal long axis (*Zhang and Jacobs, 2015*) relative to the 180° pole-to-pole shift observed in rodents (*Patel et al., 2012*), this finding implies that the recorded oscillation was also resilient to changes in recording depth. This may be the result of recording from large interlaminar macroelectrodes (see Discussion).

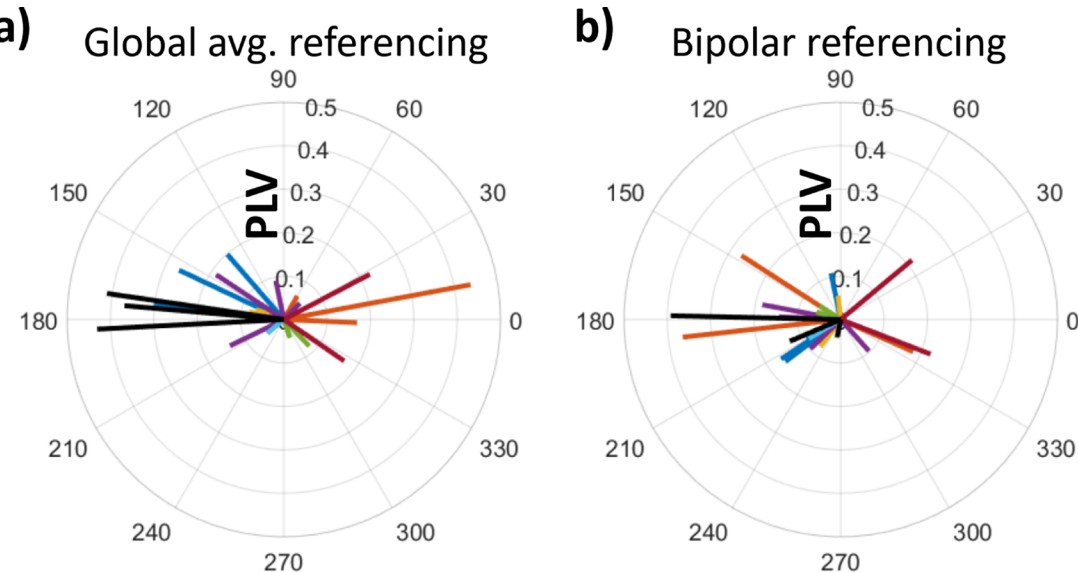

**Appendix 1—figure 4.** Distributions of phase offsets between lateral temporal and hippocampal theta were uniform across electrodes. (**a**) Distribution of theta-phase offsets between lateral temporal cortex and hippocampus under global average referencing scheme (as in *Figures 1–4*). Dashed line indicates circular-mean phase angle across electrodes. Each line represents one hippocampal electrode. Color indicates participant-of-origin for each electrode. Electrode line length indicates phase-locking value. (**b**) As (**a**), under bipolar referencing scheme wherein data from each electrode were referenced to the adjacent lateral contact on the same depth electrode.

## Hippocampal EP timing did not predict amygdala late component latency

While phase dependence of the hippocampal EP was observed most strongly during its early component (27–113 ms), phase dependence of the amygdala EP was strongest during its late component (94–256 ms). We hypothesized that the amygdala late effect may have been driven by secondary transmission from hippocampus. To investigate, we analyzed whether the amygdala late component preceded the hippocampal early component on a per-subject basis.

We analyzed data from all subjects with both amygdala and hippocampal electrodes ($n = 3$ subjects). For each subject, we computed the phase-balanced, grand average EP across electrodes for each of hippocampus and amygdala (as in Materials and methods: Theta-phase estimation). We then identified component timing on a per-subject basis, using the same procedure as described for the across-electrode dataset. For all subjects, the peak of the amygdala N2 was preceded by the peak of the hippocampal N1 for the grand average EPs (per subject timing of hippocampal N1, amygdala N2 peaks: [+34 ms,+161 ms], [+61 ms,+130 ms], [+133 ms]).

Next, we assessed whether the timing of the amygdala response tracked the timing of the hippocampal early component across trials. For each subject, we computed the mean hippocampal and amygdala EPs across electrodes for each trial. We estimated trialwise component timing using the same procedure as described for the across-electrode dataset. We then assessed the correlation between the timing of the hippocampal N1 peak and the amygdala N2 peak. This analysis was restricted to trials which passed the exclusion criteria across all relevant amygdala and hippocampal channels (see Materials and methods: sEEG recording and stimulation). No subjects showed a significant relationship between hippocampal and EP component timing (per subject component timing correlations, subject 1: $r = -0.0005$, p > 0.9; subject 2: $r = 0.007$, p = 0.9; subject 3: $r = 0.04$, p = 0.4).

