## [Editor Report]

This paper provides important evidence in humans that the phase of neuronal oscillations is a key factor that varies functional processing in the brain and that phase information can be used to modulate activity in the memory network. The authors' findings of phase-sensitive stimulation effects are quite compelling because they use direct recordings from the human brain with advanced analytical tools, thus extending work in animals. This study will likely be of widespread interest, including for researchers interested in using brain stimulation to enhance human cognition as well as scientists who probe learning and memory in simpler animals.

---

## [Decision Letter]

**Decision letter after peer review:**

Thank you for submitting your article "Human hippocampal responses to network stimulation vary with theta phase" for consideration by *eLife*. Your article has been reviewed by 3 peer reviewers, one of whom is a member of our Board of Reviewing Editors, and the evaluation has been overseen by Laura Colgin as the Senior Editor. The following individuals involved in review of your submission have agreed to reveal their identity: James Hyman (Reviewer #2); Elliot H Smith (Reviewer #3).

Essential revisions:

1. The paper should improve or correct its analyses of the phase differences between the hippocampal and neocortical contacts, because the current analyses that suggested at 90 degree phase difference do not seem supported by the data.

2. Verify that the results persist when full circular-linear models are used instead of the binned data (which leave room for inaccuracies) – see comment from reviewer 3.

3. Perform follow up analyses to ensure that the effects are not driven by the timing/order of stimulation. This could be a confound, for example, if peak versus trough stimulation trials are not evenly distributed over the course of each session.

4. Verify that the results are not driven by volume conduction that could disrupt the phase measurements.

*Reviewer #1 (Recommendations for the authors):*

– The paper should show more detail regarding how many electrodes were excluded based on the different exclusion criteria described in the Methods section. For example, "To prune excessively noisy or artifactual data, epochs were excluded according to their signal range (excluded if > 800 μV) and kurtosis (excluded if > 2 SD over channelwise mean kurtosis). Channels were excluded from analyses if < 200 epochs remained following pruning." How many channels and epochs were excluded from this procedure?

– Similarly, the #'s of excluded/included electrodes should be provided in the text at "Of these, we excluded electrodes where stimulation provoked seizure or afterdischarges during clinical testing."

– How many stimulation electrodes and hippocampus electrodes were included in the analysis in Figure 1? Similarly, how many amygdala and OFC electrodes were included in and excluded from analysis? How many electrodes are shown in Figure 10? Also, how many epochs were included for analysis for amygdala and OFC?

– It is unclear why/if the 0.5 Hz stimulation is not jittered, whereas the 1 Hz stimulation has a 0.25 second jitter.

– It would be helpful to show a time course of evoked hippocampal responses across stimulation bouts. 60 pulses were delivered per train in the 0.5 Hz setting and 1200 pulses were delivered per train in the 1 Hz setting. Given the rapid and extended delivery of pulses, the authors should show that there was no contamination from previous stimulation pulses on later evoked potentials.

– A detailed analysis of the phase estimation procedure is provided, showing a small and consistent phase angle bias of ~11 degrees. It is unclear if the authors corrected subsequent analyses based on this estimate.

– In Figure 4a, it is unclear what method is used for abolishment of oscillatory activity in the trace of phase matched stimulation-free trials. Is this done with the same method as in Figure 4b, by subtraction of average phase-matched stimulation free trials, or through a band-stop filter applied post stimulation?

– The authors suggest that late-phase dependence of evoked potentials in the amygdala may be driven by secondary transmission from hippocampus. To more strongly support this claim, it would be interesting to see a direct analysis showing hippocampal responses preceding amygdalar responses on a per subject basis.

*Reviewer #2 (Recommendations for the authors):*

My one major concern is about the state of hippocampal LFPs prior to stimulation. Presumably, patients were not in constant high-powered theta states throughout the recording session. In most comparable rodent experiments, high powered theta states are either pharmacologically controlled (in anesthetized preps) or behaviorally controlled by having animals run during stimulation periods. The authors do not mention any analysis to determine if patient's were in high-powered theta states, so presumably, all data are included. Certainly, phase can be assessed in the theta range even in the absence of clear theta in raw signals, so couldn't the variance in theta power also affect evoked potential strength. It would be nice to see a control analysis, that includes stimulation events that only occurred during periods of ongoing high-powered theta, to see how this effects the results.

While I really like the methods the authors use to assess phase lag and potentially explain the peak=higher power EP results, they authors spend a bit too much time playing coy about this explanation, while they feign that the original hypothesis was wrong. A few tweaks here and there to hedge some of the "original hypothesis was wrong" statements could go a long way to helping assure the readers take home the main points of the article.

*Reviewer #3 (Recommendations for the authors):*

Neutral questions:

– Is it worth mentioning phase coding in the introduction? Is it relevant at all that bats and humans exhibit non-oscillatory hippocampal spike-phase coding, where rodent spike-phase coding is highly oscillatory? That might change one's prior about whether the same oscillatory patterns would be visible in rodents and humans.

Major concerns:

– Why bin phase values instead of doing circular-linear regression from the outset? Finding a continuous relationship between phase and stimulation amplitude would be much more convincing. Even showing the peak-trough difference in context with the other two bins in Figure 5 would be more convincing. It seems these analyses are redundant, so in this reviewer's opinion, the best strategy would be to do circular linear analyses and show those data instead of the binned data.

– It is critical to control for timing/order of stimulation. Stimulation later could be producing smaller EPs, and if more trough stimulation trials occurred later, that could be producing the reported effect.

– The choice of a pre-determined theta frequency band for human subjects would be improved by examining a band defined by a peak in the theta range and its half width or something similar. How do you know a predefined 3-8 Hz frequency band is relevant?

– Since stimulation was applied to contacts on the same electrode as the recording contact, how can the authors be sure that volume conduction from local stimulation sites (in the lateral temporal cortex) was not affecting their results? Do the results hold with Laplacian re-referencing? This issue might explain the unexpected phase reversal that the authors mention.

---

## [Author Response]

Essential revisions:1. The paper should improve or correct its analyses of the phase differences between the hippocampal and neocortical contacts, because the current analyses that suggested at 90 degree phase difference do not seem supported by the data.

We have substantially revised the analysis and interpretation of phase differences between hippocampal and neocortical contacts. We agree that our original interpretation of a mean 90° latency was not supported given the bimodal distribution of latencies. We agree that the bimodal distribution of lateral temporal – hippocampal phase latencies may have been caused by volume conduction. In an effort to reduce volume conduction, we reanalyzed our data using bipolar re-referencing (Appendix 1-Figure 4b); however, a similar bimodal distribution persisted after re-referencing.

Analyses of phase lag thus could have been partially contaminated by volume conduction (see Essential Revisions #4), and have therefore de-emphasized this aspect of the results. Analyses of phase lag have been moved to the appendix (Appendix 1-Figure 4). We no longer incorporate findings regarding phase latency into our main conclusions. Instead, in response to editor and reviewer comments, the circular-linear analyses have been expanded and are now the main focus of the Results and Discussion sections. These analyses are indifferent to the specific phase angles relating to maximal versus minimal evoked response. The main conclusions of the revised manuscript are thus no longer related to the question of hippocampal-neocortical phase lag.

2. Verify that the results persist when full circular-linear models are used instead of the binned data (which leave room for inaccuracies) – see comment from reviewer 3.

We have reorganized the manuscript such that circular-linear models are the primary analysis. Results persist using this approach.

3. Perform follow up analyses to ensure that the effects are not driven by the timing/order of stimulation. This could be a confound, for example, if peak versus trough stimulation trials are not evenly distributed over the course of each session.

The revised manuscript includes a supplementary analysis of the effect of stimulation order (i.e. position in the stimulation session) on the distribution of stimulation phases and amplitudes (Appendix 1-Figure 2). There was no effect of stimulation order on either the uniformity or directionality of stimulation phases.

4. Verify that the results are not driven by volume conduction that could disrupt the phase measurements.

This issue applies to the analysis of phase lag between hippocampus and neocortex. As indicated above, we no longer emphasize this analysis given the potential issue of volume conduction (see response to Essential Revisions #1) and in general have de-emphasized the relevance of absolute phase values to our results. In the revised manuscript, we include an analysis of the effect of an alternative referencing scheme selected to reduce volume conduction (bipolar re-referencing) on the phase latencies between hippocampus and the stimulating contacts (Appendix 1-Figure 4b). It is important to note that issues concerning volume conduction would not be expected to impact the main reported finding of 180-degree hippocampal theta phase separation between minimal and maximal response amplitudes. We also note that the large distance between stimulating and recording contacts should reduce volume conduction effects.

Reviewer #1 (Recommendations for the authors):– The paper should show more detail regarding how many electrodes were excluded based on the different exclusion criteria described in the Methods section. For example, "To prune excessively noisy or artifactual data, epochs were excluded according to their signal range (excluded if > 800 μV) and kurtosis (excluded if > 2 SD over channelwise mean kurtosis). Channels were excluded from analyses if < 200 epochs remained following pruning." How many channels and epochs were excluded from this procedure?

We now provide information on the numbers of channels included/excluded and the number of epochs included/excluded for hippocampus and control regions (Materials and methods: sEEG recording and stimulation).

– Similarly, the #'s of excluded/included electrodes should be provided in the text at "Of these, we excluded electrodes where stimulation provoked seizure or after discharges during clinical testing."

For clarity, we now include the total number of included stimulating electrodes in Materials and methods: sEEG recording and stimulation (n=16). Electrodes where stimulation provoked seizure or after discharge during clinical testing were excluded from consideration and were not assessed by the research team. Given the large volume of electrodes stimulated over the course of clinical testing, the research team only noted the lateral temporal contacts that were possibilities for use in the stimulation experiment. The research team selected from among those pairs using the evaluation procedure described in Materials and methods: sEEG recording and stimulation. This procedure did not evoke afterdischarge or seizure in any evaluated electrode pair.

– How many stimulation electrodes and hippocampus electrodes were included in the analysis in Figure 1?

14 neocortical stimulating electrodes and 18 hippocampal recording electrodes were included in the electrode visualization in Figure 1a. We now include this information in the figure caption. We now also emphasize in the figure caption that imaging was unavailable in one subject and they were therefore excluded from this visualization, as this information was previously described only in Materials and methods. All analyses include the full dataset of 16 stimulating electrodes and 23 hippocampal recording electrodes.

Similarly, how many amygdala and OFC electrodes were included in and excluded from analysis? Also, how many epochs were included for analysis for amygdala and OFC?

As was the case for hippocampal electrodes, the revised manuscript provides channel inclusion/exclusion and epoch inclusion/exclusion counts in Materials and methods: sEEG recording and stimulation.

How many electrodes are shown in Figure 10?

9 amygdala electrodes and 22 orbitofrontal electrodes (i.e., the full analysis set) are shown in Figure 5. We now emphasize this information in the figure caption as well as in the Results section.

– It is unclear why/if the 0.5 Hz stimulation is not jittered, whereas the 1 Hz stimulation has a 0.25 second jitter.

We now explain in Materials and methods that jitter was applied to the 1-Hz stimulation in order to enable data analysis for a secondary experiment.

– It would be helpful to show a time course of evoked hippocampal responses across stimulation bouts. 60 pulses were delivered per train in the 0.5 Hz setting and 1200 pulses were delivered per train in the 1 Hz setting. Given the rapid and extended delivery of pulses, the authors should show that there was no contamination from previous stimulation pulses on later evoked potentials.

The revised manuscript addresses the effect of stimulation order on evoked amplitude (as well as on stimulation phase distribution) in the new appendix (Appendix 1: Order of stimulation pulses had no effect on phase distribution). There were no significant effects of stimulation order.

– A detailed analysis of the phase estimation procedure is provided, showing a small and consistent phase angle bias of ~11 degrees. It is unclear if the authors corrected subsequent analyses based on this estimate.

We did not rebin phase angles according to the estimated bias from the phase estimation procedure in our main analyses. Our revised manuscript reports the effects of performing this correction on our main analyses (see Appendix 1: Assessment of phase estimation approach). We found no change in the outcomes of our binned analyses of EP component amplitude. This procedure has no effect on the continuous (circular-linear) analyses.

– In Figure 4a, it is unclear what method is used for abolishment of oscillatory activity in the trace of phase matched stimulation-free trials. Is this done with the same method as in Figure 4b, by subtraction of average phase-matched stimulation free trials, or through a band-stop filter applied post stimulation?

The phase-matched, stimulation-free trials shown in the revised manuscript in Figures 3a,c and Figure 5c,d,g,h do not have any abolishment of oscillatory activity applied. Rather, these traces show what the underlying, non-evoked activity looks like when revealed by phase-sorting. We abolish the oscillatory activity from the original EP by subtracting the average phase-matched stimulation-free trial from the original EP. For example, the black (peak) trace in Figure 3b is the result of subtracting the phase-matched, stimulation-free trace (red) from the raw EP trace (black) in Figure 3a, top panel. The revised manuscript includes retooled visualization, figure layout, and caption in an effort to clarify this point.

– The authors suggest that late-phase dependence of evoked potentials in the amygdala may be driven by secondary transmission from hippocampus. To more strongly support this claim, it would be interesting to see a direct analysis showing hippocampal responses preceding amygdalar responses on a per subject basis.

The revised manuscript includes an analysis of per-subject latency between the hippocampal EP and the amygdala late response. While the hippocampus N1 consistently preceded the amygdala N2, we found no correlation across trials between the onset of the hippocampus N1 and the amygdala N2 (see Appendix 1: Amygdala late-component latency does not relate to hippocampal EP timing). We have updated our results and Discussion sections accordingly.

Reviewer #2 (Recommendations for the authors):My one major concern is about the state of hippocampal LFPs prior to stimulation. Presumably, patients were not in constant high-powered theta states throughout the recording session. In most comparable rodent experiments, high powered theta states are either pharmacologically controlled (in anesthetized preps) or behaviorally controlled by having animals run during stimulation periods. The authors do not mention any analysis to determine if patient's were in high-powered theta states, so presumably, all data are included. Certainly, phase can be assessed in the theta range even in the absence of clear theta in raw signals, so couldn't the variance in theta power also affect evoked potential strength. It would be nice to see a control analysis, that includes stimulation events that only occurred during periods of ongoing high-powered theta, to see how this effects the results.

Due to contamination by stimulation and the evoked response, we were unable to analyze theta bouts online (i.e., on a per-trial basis). We instead investigated bout incidence during the offline, pre-stimulation period. There was no relationship between offline bouts and the phase-dependence of the hippocampal evoked response (Appendix 1: Offline theta bout incidence does not impact periodicity of the evoked response). We now note in the Discussion section that this is an important direction for future research, perhaps by cuing stimulation based on online assessment of theta power.

While I really like the methods the authors use to assess phase lag and potentially explain the peak=higher power EP results, they authors spend a bit too much time playing coy about this explanation, while they feign that the original hypothesis was wrong. A few tweaks here and there to hedge some of the "original hypothesis was wrong" statements could go a long way to helping assure the readers take home the main points of the article.

We believe that the revised manuscript’s emphasis on full circular-linear models helps resolve this issue. In particular, we focus on the finding of 180 degree phase separation between minimal and maximal receptivity states, thus emphasizing our finding’s homology to previous findings in rodent models.

Reviewer #3 (Recommendations for the authors):Neutral questions:– Is it worth mentioning phase coding in the introduction? Is it relevant at all that bats and humans exhibit non-oscillatory hippocampal spike-phase coding, where rodent spike-phase coding is highly oscillatory? That might change one's prior about whether the same oscillatory patterns would be visible in rodents and humans.

In the revised Introduction, we now directly address differences in characteristics and functional relevance of the theta oscillation across species.

Major concerns:– Why bin phase values instead of doing circular-linear regression from the outset? Finding a continuous relationship between phase and stimulation amplitude would be much more convincing. Even showing the peak-trough difference in context with the other two bins in Figure 5 would be more convincing. It seems these analyses are redundant, so in this reviewer's opinion, the best strategy would be to do circular linear analyses and show those data instead of the binned data.

We now use the circular-linear regressions from the outset (see Essential Revisions #2). We have retained the binning analysis secondarily.

– It is critical to control for timing/order of stimulation. Stimulation later could be producing smaller EPs, and if more trough stimulation trials occurred later, that could be producing the reported effect.

The revised manuscript includes a supplementary analysis of the effect of stimulation order (see Essential Revisions #3). We did not find effects of stimulation order on the distribution of stimulation phase angles.

– The choice of a pre-determined theta frequency band for human subjects would be improved by examining a band defined by a peak in the theta range and its half width or something similar. How do you know a predefined 3-8 Hz frequency band is relevant?

We now provide analyses of power spectra in hippocampus to verify the presence of significant narrowband oscillations within the 3-8 Hz frequency band (see revised Figure 1c). In particular, we detected consistent oscillations around 5.3 Hz and 6.3 Hz. All analyzed contacts were found to have at least one narrowband peak in the 3-8 Hz frequency range.

– Since stimulation was applied to contacts on the same electrode as the recording contact, how can the authors be sure that volume conduction from local stimulation sites (in the lateral temporal cortex) was not affecting their results? Do the results hold with Laplacian re-referencing? This issue might explain the unexpected phase reversal that the authors mention.

Volume conduction may have contaminated our findings in regard to the hippocampal-neocortical latency. As such, we have removed those analyses from our interpretation in the Results and Discussion sections, moved the phase-latency analysis to the appendix, and have generally de-emphasized the absolute phase values relating to minimal and maximal evoked response (see Essential Revisions #1 and #4). Laplacian re-referencing was not possible for many hippocampal contacts which were at the end of their depth electrode lead. The revised manuscript, we include an analysis of the effect of bipolar re-referencing, which did not eliminate the bimodal phase-lag distribution observed in the original reference scheme (see Essential Revisions #1). However, we note that volume conduction would not be expected to impact the primary results of circular-linear phase to evoked-amplitude relationship and 180-degree hippocampal theta phase separation in minimal vs. maximal response to stimulation.